# Pre-trained Adversarial Perturbations

**Yuanhao Ban**[1,2*], **Yinpeng Dong**[1,3†]

[1] Department of Computer Science & Technology, Institute for AI, BNRist Center,
Tsinghua-Bosch Joint ML Center, THBI Lab, Tsinghua University
[2] Department of Electronic Engineering, Tsinghua University     [3] RealAI
banyh19@mails.tsinghua.edu.cn, dongyinpeng@mail.tsinghua.edu.cn

## Abstract

Self-supervised pre-training has drawn increasing attention in recent years due to its superior performance on numerous downstream tasks after fine-tuning. However, it is well-known that deep learning models lack the robustness to adversarial examples, which can also invoke security issues to pre-trained models, despite being less explored. In this paper, we delve into the robustness of pre-trained models by introducing Pre-trained Adversarial Perturbations (PAPs), which are universal perturbations crafted for the pre-trained models to maintain the effectiveness when attacking fine-tuned ones without any knowledge of the downstream tasks. To this end, we propose a Low-Level Layer Lifting Attack (L4A) method to generate effective PAPs by lifting the neuron activations of low-level layers of the pre-trained models. Equipped with an enhanced noise augmentation strategy, L4A is effective at generating more transferable PAPs against fine-tuned models. Extensive experiments on typical pre-trained vision models and ten downstream tasks demonstrate that our method improves the attack success rate by a large margin compared with state-of-the-art methods.

## 1  Introduction

Large-scale pre-trained models [40, 13] have recently achieved unprecedented success in a variety of fields, e.g., natural language processing [21, 27, 1], computer vision [2, 16, 17]. A large amount of work proposes sophisticated self-supervised learning algorithms, enabling the pre-trained models to extract useful knowledge from large-scale unlabeled datasets. The pre-trained models consequently facilitate downstream tasks through transfer learning or fine-tuning [37, 50, 12]. Nowadays, more practitioners without sufficient computational resources or training data tend to fine-tune the publicly available pre-trained models on their own datasets. Therefore, it has become an emerging trend to adopt the paradigm of pre-training to fine-tuning rather than training from scratch [13].

Despite the excellent performance of deep learning models, they are incredibly vulnerable to adversarial examples [44, 11], which are generated by adding small, human-imperceptible perturbations to natural examples, but can make the target model output erroneous predictions. Adversarial examples also exhibit an intriguing property called *transferability* [44, 26, 32], which means that the adversarial perturbations generated for one model or a set of images can remain adversarial for others. For example, a universal adversarial perturbation (UAP) [32] can be generated for the entire distribution of data samples, demonstrating excellent cross-data transferability. Other work [26, 7, 47, 8, 34] has revealed that adversarial examples have high cross-model and cross-domain transferability, making *black-box attacks* practical without any knowledge of the target model or even the training data. However, much less effort has been devoted to exploring the adversarial robustness of pre-trained models. As these models have been broadly studied and deployed in various real-world applications,

---

*This work was done when Yuanhao Ban was intern at RealAI, Inc; †Corresponding author.

36th Conference on Neural Information Processing Systems (NeurIPS 2022).

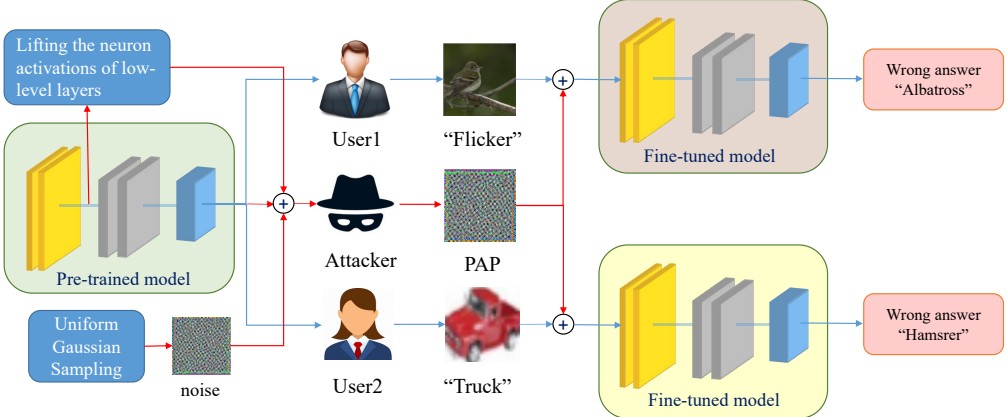

Figure 1: A demonstration of pre-trained adversarial perturbations (PAPs): An attacker first downloads pre-trained weights on the Internet and generates a PAP by lifting the neuron activations of low-level layers of the pre-trained models. We adopt a data augmentation technique called uniform Gaussian sampling to improve the transferability of PAP. When users fine-tune the pre-trained models to complete downstream tasks, the attacker can add the PAP to the input of the fine-tuned models to cheat them without knowing the specific downstream tasks.

it is of significant importance to identify their weaknesses and evaluate their robustness, especially concerning the pre-training to the fine-tuning procedure.

In this paper, we introduce **Pre-trained Adversarial Perturbations (PAPs)**, a new kind of universal adversarial perturbations designed for pre-trained models. Specifically, a PAP is generated for a pre-trained model to effectively fool any downstream model obtained by fine-tuning the pre-trained one, as illustrated in Fig. 1. It works under a quasi-black-box setting where the downstream task, dataset, and fine-tuned model parameters are all unavailable. This attack setting is more suitable for the pre-training to the fine-tuning procedure since many pre-trained models are publicly available, and the adversary may generate PAPs before the pre-trained model has been fine-tuned. Although there are many methods [7, 47] proposed for improving the transferability, they do not consider the specific characteristics of the pre-training to the fine-tuning procedure, limiting their *cross-finetuning transferability* in our setting.

To generate more effective PAPs, we propose a **Low-Level Layer Lifting Attack (L4A)** method, which aims to lift the feature activations of low-level layers. Motivated by the finding that the lower the level of the model's layer is, the less its parameters change during fine-tuning, we generate PAPs to destroy the low-level feature representations of pre-trained models, making the attacking effects better reserved after fine-tuning. To further alleviate the overfitting of PAPs to the source domain, we improve L4A with a noise augmentation technique. We conduct extensive experiments on typical pre-trained vision models [2, 17] and ten downstream tasks. The evaluation results demonstrate that our method achieves a higher attack success rate on average compared with the alternative baselines.

## 2   Related work

**Self-supervised learning.** Self-supervised learning (SSL) enables learning from unlabeled data. To achieve this, early approaches utilize hand-crafted pretext tasks, including colorization [53], rotation prediction [10], position prediction [36], and Selfie [45]. Another approach for SSL is contrastive learning [25, 39, 2, 22], which aims to map the input image to the feature space and minimize the distance between similar ones while keeping dissimilar ones far away from each other. In particular, a similar sample is retrieved by applying appropriate data augmentation techniques to the original one, and the versions of different samples are viewed as dissimilar pairs.

**Adversarial examples.** With the knowledge of the structure and parameters of a model, many algorithms [24, 31, 30, 38] successfully fool the target model in a white-box manner. An intriguing property of adversarial examples is their good transferability [26, 32]. The universal adversarial perturbations [32] demonstrate good cross-data transferability by optimizing under a distribution of

data samples. The cross-model transferability has also been extensively studied [7, 47, 8], enabling the attack on black-box models without any knowledge of their internal working mechanisms.

**Robustness of the pre-training to fine-tuned procedure.** Due to the popularity of pre-trained models, a lot of works [43, 49, 4] study the robustness of this setting. Among them, Dong et al. [6] propose a novel adversarial fine-tuning method in an information-theoretical way to retain robust features learned from the pre-trained model. Jiang et al. [20] integrate adversarial samples into the pre-training procedure to defend against attacks. Fan et at. [9] adopt Clusterfit [48] to generate pseudo-label data and later use them for training the model in a supervised way, which improves the robustness of the pre-trained model. The main difference between our work and theirs is that we consider the problem from an attacker's perspective.

## 3 Methodology

In this section, we first introduce the notations and the problem formulation of the Pre-trained Adversarial Perturbations (PAPs). Then, we detail the Low-Level Layer Lifting Attack (L4A) method.

### 3.1 Notations and problem formulation

Let $f_{\boldsymbol{\theta}}$ denote a pre-trained model for feature extraction with parameters $\boldsymbol{\theta}$. It takes an image $\mathbf{x} \in \mathcal{D}_p$ as input and outputs a feature vector $\mathbf{v} \in \mathcal{X}$, where $\mathcal{D}_p$ and $\mathcal{X}$ refer to the pre-training dataset and feature space, respectively. We denote $f_{\boldsymbol{\theta}}^k(\mathbf{x})$ as the $k$-th layer's feature map of $f_{\boldsymbol{\theta}}$ for an input image $\mathbf{x}$. In the pre-training to fine-tuning paradigm, a user fine-tunes the pre-trained model $f_{\boldsymbol{\theta}}$ using a new dataset $\mathcal{D}_t$ of the downstream task and finally gets a fine-tuned model $f_{\boldsymbol{\theta}'}$ with updated parameters $\boldsymbol{\theta}'$. Then, let $f_{\boldsymbol{\theta}'}(\mathbf{x})$ be the predicted probability distribution of an image $\mathbf{x}$ over the classes of $\mathcal{D}_t$, and $F_{\boldsymbol{\theta}'}(\mathbf{x}) = \arg\max f_{\boldsymbol{\theta}'}(\mathbf{x})$ be the final classification result.

In this paper, we introduce **Pre-trained Adversarial Perturbations (PAPs)**, which are generated for the pre-trained model $f_{\boldsymbol{\theta}}$, but can effectively fool fine-tuned models $f_{\boldsymbol{\theta}'}$ on downstream tasks. Formally, a PAP is a universal perturbation $\boldsymbol{\delta}$ within a small budget $\epsilon$ crafted by $f_{\boldsymbol{\theta}}$ and $\mathcal{D}_p$, such that $F_{\boldsymbol{\theta}'}(\mathbf{x} + \boldsymbol{\delta}) \neq F_{\boldsymbol{\theta}'}(\mathbf{x})$ for most of the instances belonging to the fine-tuning dataset $\mathcal{D}_t$. This can be formulated as the following optimization problem:

$$\max_{\boldsymbol{\delta}} \mathbb{E}_{\mathbf{x} \sim \mathcal{D}_t}[F_{\boldsymbol{\theta}'}(\mathbf{x}) \neq F_{\boldsymbol{\theta}'}(\mathbf{x} + \boldsymbol{\delta})], \text{ s.t. } \|\boldsymbol{\delta}\|_p \leq \epsilon \text{ and } \mathbf{x} + \boldsymbol{\delta} \in [0, 1], \tag{1}$$

where $\| \cdot \|_p$ denotes the $\ell_p$ norm and we take the $\ell_\infty$ norm in this work. There exist some works related to the universal perturbations, such as the universal adversarial perturbation (UAP) [32] and the fast feature fool (FFF) [33], as detailed below.

**UAP**: Given a classifier $f$ and its dataset $\mathcal{D}$, the UAP tries to generate a perturbation $\boldsymbol{\delta}$ that can fool the model on most of the instances from $\mathcal{D}$, which is usually solved by an iterative method. Every time sampling an image $\mathbf{x}$ from the dataset $\mathcal{D}$, the attacker computes the minimal perturbation $\boldsymbol{\zeta}$ that sends $\mathbf{x} + \boldsymbol{\delta}$ to the decision boundary by Eq. (2) and then adds it into $\boldsymbol{\delta}$.

$$\boldsymbol{\zeta} \leftarrow \arg\min_{\mathbf{r}} \|\mathbf{r}\|_2, \text{ s.t. } F(\mathbf{x} + \boldsymbol{\delta} + \mathbf{r}) \neq F(\mathbf{x}). \tag{2}$$

**FFF**: It aims to produce maximal spurious activations at each layer. To achieve this, FFF starts with a random $\boldsymbol{\delta}$ and solves the following problem:

$$\min_{\boldsymbol{\delta}} -\log\left(\prod_{i=0}^{K} \bar{l}_i(\boldsymbol{\delta})\right), \text{ s.t. } \|\boldsymbol{\delta}\|_p \leq \epsilon. \tag{3}$$

where $\bar{l}_i(\boldsymbol{\delta})$ is the mean of the output tensor at layer $i$.

### 3.2 Our design

However, these attacks show limited cross-finetuning transferability in our problem setting due to ignorance of the fine-tuning procedure. Two challenges are degenerating the performance.

- **Fine-tuning Deviation.** The parameters of the model could change a lot during fine-tuning. As a result, the generated adversarial samples may perform well in the feature space of the pre-trained model but fail in the fine-tuned ones.

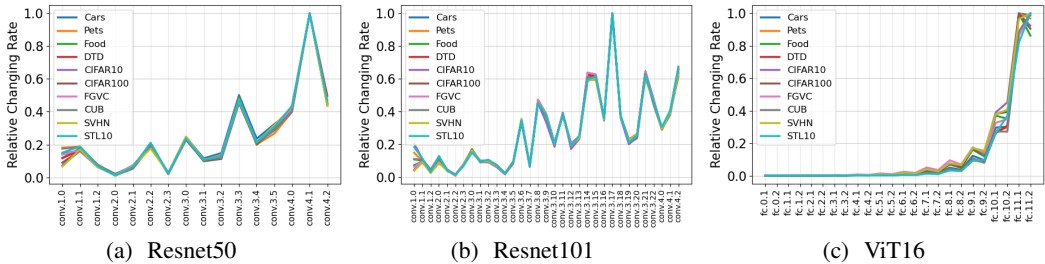

| (a) Resnet50 | (b) Resnet101 | (c) ViT16 |

Figure 2: The ordinate represents the Frobenius norm of the difference between the parameters of the fine-tuned model and its corresponding pre-trained model, which is scaled into a range from 0 to 1 for easy comparison. The abscissa represents the level of the layer. Note that Resnet50 and Resnet101 [14] are pre-trained by SimCLRv2 [2], and ViT16 [46] is pre-trained by MAE [17].

- **Datasets Deviation.** The statistics (i.e., mean and standard deviation) of different datasets can vary a lot. Only using the pre-training dataset with the fixed statistics to generate adversarial samples may suffer a performance drop.

To alleviate the negative effect of the above issues, we propose a **Low-Level Layer Lifting Attack (L4A)** method equipped with a **uniform Gaussian sampling** strategy.

**Low-Level Layer Lifting Attack (L4A).** Our method is motivated by the findings in Fig. 2 that the higher the level of the layers, the more their parameters change during fine-tuning. This is also consistent with the knowledge that the low-level convolutional layer acts as an edge detector that extracts low-level features like edges and textures and has little high-level semantic information [37, 50]. Since images from different datasets share the same low-level features, the parameters of these layers can be preserved during fine-tuning. In contrast, the attack algorithms based on the high-level layers or the scores predicted by the model may not transfer well in such a cross-finetuning setting, as the feature spaces of high-level layers are easily distorted during fine-tuning. The basic method of L4A can be formulated as the following problem:

$$\min_{\boldsymbol{\delta}} L_{base}(f_{\boldsymbol{\theta}}, \mathbf{x}, \boldsymbol{\delta}) = -\mathbb{E}_{\mathbf{x}\sim D_p}\left[\|f_{\boldsymbol{\theta}}^k(\mathbf{x}+\boldsymbol{\delta})\|_F^2\right], \tag{4}$$

where $\|\cdot\|_F$ denotes the Frobenius norm of the input tensor. In our experiments, we find the lower the layer, the better it performs, so we choose the first layer as default, such that $k = 1$. As Eq. (4) is usually a sophisticated non-convex optimization problem, we solve it using stochastic gradient descent method.

We also find that fusing the adversarial loss of the consecutive low-level layers can boost the performance, which gives L4A$_{fuse}$ method as solving:

$$\min_{\boldsymbol{\delta}} L_{fuse}(f_{\boldsymbol{\theta}}, \mathbf{x}, \boldsymbol{\delta}) = -\mathbb{E}_{\mathbf{x}\sim D_p}\left[\|f_{\boldsymbol{\theta}}^{k_1}(\mathbf{x}+\boldsymbol{\delta})\|_F^2 + \lambda \cdot \|f_{\boldsymbol{\theta}}^{k_2}(\mathbf{x}+\boldsymbol{\delta})\|_F^2\right], \tag{5}$$

where $f_{\boldsymbol{\theta}}^{k_1}(\mathbf{x}+\boldsymbol{\delta})$ and $f_{\boldsymbol{\theta}}^{k_2}(\mathbf{x}+\boldsymbol{\delta})$ refers to the $k_1$-th and $k_2$-th layers' feature maps of $f_{\boldsymbol{\theta}}$ respectively, $\lambda$ is a balancing hyperparameter. We set $k_1 = 1$ and $k_2 = 2$ as default.

**Uniform Gaussian Sampling.** Nowadays, most state-of-the-art networks apply batch normalization [19] to input images for better performance. Thus, the datasets' statistics become an essential factor for training. As shown in Fig. 3, the distribution of the downstream datasets can vary significantly compared to that of the pre-training dataset. However, traditional data augmentation techniques [51, 18] are limited to the pre-training domain and cannot alleviate the problem. Thus, we propose sampling Gaussian noises with various means and deviations to avoid overfitting. Combining the base loss using the pre-training dataset and the new loss using uniform Gaussian noises gives the L4A$_{ugs}$

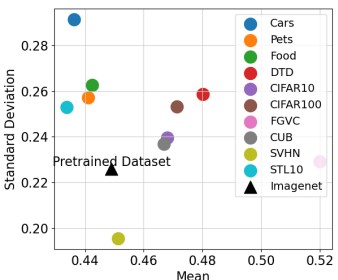

Figure 3: Datasets' statistics.

Table 1: The attack success rate (%) of various attack methods against **Resnet101** pre-trained by **SimCLRv2**. Note that C10 stands for CIFAR10 and C100 stands for CIFAR100.

| ASR | Cars | Pets | Food | DTD | FGVC | CUB | SVHN | C10 | C100 | STL10 | AVG |
|---|---|---|---|---|---|---|---|---|---|---|---|
| FFF$_{no}$ | 43.81 | 38.62 | 49.95 | 63.24 | 85.57 | 48.38 | 12.55 | 8.53 | 77.74 | 57.11 | 48.55 |
| FFFF$_{mean}$ | 33.93 | 31.37 | 41.77 | 52.66 | 78.94 | 45.00 | **14.85** | 14.42 | 72.59 | 56.66 | 44.22 |
| FFF$_{one}$ | 31.87 | 29.74 | 39.25 | 46.92 | 74.17 | 43.87 | 9.24 | 11.77 | 65.61 | 50.21 | 40.26 |
| DR | 36.28 | 35.54 | 47.43 | 47.45 | 75.00 | 44.15 | 12.05 | 21.35 | 65.39 | 41.65 | 42.63 |
| SSP | 32.89 | 30.50 | 43.12 | 45.85 | 82.57 | 45.55 | 8.69 | 11.66 | 65.80 | 40.91 | 40.75 |
| ASV | 60.75 | 19.84 | 36.33 | 56.22 | 84.16 | 55.82 | 7.11 | 7.29 | 58.10 | 80.89 | 46.64 |
| UAP | 48.70 | 36.55 | 60.80 | 63.40 | 76.06 | 52.64 | 8.46 | 8.53 | 52.35 | 31.15 | 43.86 |
| UAPEPGD | 94.12 | 66.66 | 61.30 | 72.55 | 70.34 | 82.72 | 13.88 | **61.65** | 20.04 | 50.13 | 59.34 |
| L4A$_{base}$ | 94.07 | 61.57 | 71.23 | 69.20 | **96.28** | 81.07 | 11.70 | 12.68 | 80.57 | 90.49 | 66.89 |
| L4A$_{fuse}$ | 90.98 | 88.53 | **80.65** | 74.31 | 93.79 | 91.23 | 11.40 | 17.40 | **80.98** | 89.69 | 67.10 |
| L4A$_{ugs}$ | **94.24** | **94.99** | 78.28 | **77.23** | 92.92 | **91.77** | **11.40** | 14.60 | 76.50 | **90.05** | **72.20** |

method as follows:

$$\min_{\boldsymbol{\delta}} L_{ugs}(f_{\boldsymbol{\theta}}, \mathbf{x}, \boldsymbol{\delta}) = -\mathbb{E}_{\boldsymbol{\mu},\boldsymbol{\sigma},\mathbf{n}_0 \sim N(\boldsymbol{\mu},\boldsymbol{\sigma})} \left\{ \mathbb{E}_{\mathbf{x}\sim\mathcal{D}_p} \left[ \|f_{\boldsymbol{\theta}}^k(\mathbf{x}+\boldsymbol{\delta})\|_F^2 + \lambda \cdot \|f_{\boldsymbol{\theta}}^k(\mathbf{n}_0+\boldsymbol{\delta})\|_F^2 \right] \right\}, \quad (6)$$

where $\boldsymbol{\mu}$ and $\boldsymbol{\sigma}$ are drawn from the uniform distribution $U(\boldsymbol{\mu}_l, \boldsymbol{\mu}_h)$ and $U(\boldsymbol{\sigma}_l, \boldsymbol{\sigma}_h)$, respectively, and $\boldsymbol{\mu}_l, \boldsymbol{\mu}_h, \boldsymbol{\sigma}_l, \boldsymbol{\sigma}_r$ are four hyperparameters.

# 4   Experiments

We provide some experimental results in this section. More results can be found in Appendix. Our code is publicly available at https://github.com/banyuanhao/PAP.

## 4.1   Settings

**Pre-training methods.** SimCLR [2, 3] uses the Resnet [14] backbone and pre-trains the model by contrastive learning. We download pre-trained parameters of Resnet50 and Resnet101[1] to evaluate the generalization ability of our algorithm on different architectures. We also adopt MOCO [15] with the backbone of Resnet50[2]. Besides convolutional neural networks, transformers [46] attract much attention nowadays for their competitive performance. Based on transformers and masked image modeling, MAE [17] becomes a good alternative for pre-training. We adopt the pre-trained ViT-base-16 model[3]. Moreover, vision-language pre-trained models are gaining popularity these days. Thus we also choose CLIP [41][4] for our study. We report the results of SimCLR and MAE in Section 4.2. More results on CLIP and MOCO can be found in Appendix A.1.

**Datasets and Pre-processing.** We adopt the ILSVRC 2012 dataset [42] to generate PAPs, which are also used to pre-train the models. We mainly evaluate PAPs on image classification tasks, which are the same as the settings of SimCLRv2. Ten fine-grained and coarse-grained datasets are used to test the cross-finetuning transferability of the generated PAPs. We load these datasets from torchvision (Details in AppendixD). Before feeding the images to the model, we resize them to $256 \times 256$ and then center crop them into $224 \times 224$.

**Compared methods.** We choose UAP [32] to test whether image-agnostic attacks also bear good cross-finetuning transferability. Since UAP needs final classification predictions of the inputs, we fit a linear head on the pre-trained feature extractor. Furthermore, by integrating the moment term into the iterative method, UAPEPGD [5] is believed to enhance cross-model transferability. Thus, we adopt UAPEPGD to study the connection between cross-model and cross-finetuning transferability. As our algorithm is based on the feature level, other feature attacks (including FFF [33], ASV [23], DR [28], SSP [35]) are chosen for comparison.

---

[1]https://github.com/google-research/simclr
[2]https://dl.fbaipublicfiles.com/moco/
[3]https://github.com/facebookresearch/mae
[4]https://github.com/openai/CLIP

Table 2: The attack success rate (%) of various attack methods against **Resnet50** pre-trained by **SimCLRv2**. Note that C10 stands for CIFAR10, and C100 stands for CIFAR100.

| ASR | Cars | Pets | Food | DTD | FGVC | CUB | SVHN | C10 | C100 | STL10 | AVG |
|---|---|---|---|---|---|---|---|---|---|---|---|
| $FFF_{no}$ | 26.91 | 30.83 | 43.28 | 48.99 | 41.30 | 38.23 | 79.00 | 68.50 | 44.67 | 16.95 | 43.86 |
| $FFF_{mean}$ | 36.75 | 33.88 | 45.26 | 50.15 | 53.13 | 77.22 | 52.02 | 82.41 | 68.10 | 22.11 | 52.10 |
| $FFF_{one}$ | 37.88 | 35.30 | 52.79 | 59.52 | 59.62 | 57.04 | 80.33 | 75.40 | 53.58 | 18.31 | 52.98 |
| DR | 38.64 | 34.42 | 50.04 | 45.53 | 39.80 | 75.67 | 47.88 | 76.05 | 60.57 | 13.98 | 48.26 |
| SSP | 41.70 | 43.94 | 50.83 | 48.78 | 47.67 | 82.39 | 48.38 | 86.95 | 66.23 | 19.56 | 53.64 |
| ASV | 74.47 | 36.93 | 45.85 | 73.51 | 64.89 | 92.29 | 73.16 | 45.14 | 53.60 | 22.02 | 58.19 |
| UAP | 44.86 | 46.47 | 64.67 | 65.53 | 49.63 | 82.32 | 52.00 | 79.63 | 46.46 | 19.99 | 55.16 |
| UAPEPGD | 66.29 | 66.58 | **81.11** | 69.52 | 87.91 | 59.07 | 69.16 | **87.84** | 68.26 | 37.12 | 69.28 |
| $LLLL_{base}$ | 94.86 | 56.30 | 61.31 | 75.37 | 67.61 | 94.87 | 81.45 | 68.25 | 77.04 | 34.56 | 66.89 |
| $L4A_{fuse}$ | 96.00 | 59.80 | 65.00 | 77.93 | 69.39 | **95.02** | 85.05 | 64.41 | 76.29 | 37.54 | 72.64 |
| $L4A_{ugs}$ | **96.13** | **79.15** | 74.87 | **82.18** | **78.73** | 94.45 | **95.29** | 55.03 | **77.10** | **45.09** | **77.80** |

Table 3: The attack success rate (%) of various attack methods against **ViT16** pre-trained by **MAE**. Note that C10 stands for CIFAR10 and C100 stands for CIFAR100.

| ASR | Cars | Pets | Food | DTD | FGVC | CUB | SVHN | C10 | C100 | STL10 | AVG |
|---|---|---|---|---|---|---|---|---|---|---|---|
| $FFF_{no}$ | 64.31 | 88.21 | 95.04 | 88.18 | 81.91 | 92.94 | 76.10 | 49.48 | 79.83 | 60.91 | 77.69 |
| $FFF_{mean}$ | 40.39 | 67.54 | 54.10 | 61.38 | 71.47 | 73.39 | **92.96** | 86.88 | 94.55 | 67.90 | 71.06 |
| $FFF_{one}$ | 48.36 | 77.89 | 60.06 | 64.04 | 75.67 | 74.09 | 92.33 | 86.13 | 94.48 | 70.40 | 74.35 |
| DR | 37.02 | 23.84 | 59.54 | 44.73 | 28.01 | 10.41 | 14.30 | 16.66 | 14.12 | 21.54 | 27.02 |
| SSP | 44.15 | 73.31 | 85.42 | 72.82 | 52.57 | 63.10 | 52.45 | 27.94 | 25.32 | 36.90 | 53.40 |
| ASV | 38.46 | 10.17 | 37.49 | 48.31 | 29.14 | 4.97 | 8.41 | 17.04 | 11.34 | 21.14 | 22.64 |
| UAP | 62.71 | 58.90 | 89.90 | 74.92 | 44.69 | 39.56 | 47.65 | 47.77 | 33.80 | 51.70 | 55.16 |
| UAPEPGD | 63.67 | 73.09 | 96.22 | 76.69 | 57.78 | 73.37 | 79.84 | 45.89 | 47.21 | 55.79 | 66.95 |
| $L4A_{base}$ | 87.66 | 89.98 | 98.96 | **99.10** | 99.33 | 84.06 | 86.99 | 98.62 | 97.08 | 98.25 | 94.00 |
| $L4A_{fuse}$ | 83.24 | 89.57 | 98.87 | 98.77 | 98.36 | **93.60** | 89.85 | 98.64 | 95.72 | 97.53 | 94.42 |
| $L4A_{ugs}$ | **96.49** | **90.00** | **98.97** | 98.89 | **99.48** | 84.01 | 89.56 | **99.43** | **97.27** | **98.96** | **95.30** |

**Default settings and Metric.** Unless otherwise specified, we choose a batch size of 16 and a step size of 0.0002. All the perturbations should be within the bound of 0.05 under the $\ell_\infty$ norm. We evaluate the perturbations at the iterations of $1,000, 5,000, 30,000,$ and $60,000$, and report the best performance. We show the results in the measure of attack success rates (ASR), representing the classification error rate on the whole testing dataset after adding the perturbations to the legitimate images.

## 4.2 Main results

We craft pre-trained adversarial perturbations (PAPs) for three pre-trained models (i.e., Resnet50 by SimCLRv2, Resnet101 by SimCLRv2, ViT16 by MAE) and evaluate the attack success rates on ten downstream tasks. The results are shown in Table 1, Table 2, and Table 3, respectively. Note that the first seven datasets are fine-grained, while the last three are coarse-grained ones. We mark the best results for each dataset in **bold**, and the best baseline in blue. We highlight the results of $L4A_{ugs}$ in red to emphasize that the L4A attack equipped with Uniform Gaussian Sampling shows great *cross-finetuning transferability* and performs best.

A quick glimpse shows that our proposed methods outperform all the baselines by a large margin. For example, as can be seen from Table 1, if the target model is Resnet101 pre-trained by SimCLRv2, the best competitor $FFF_{mean}$ achieves an average attack success rate of 59.34%, while the villain $L4A_{base}$ can lift it up to **66.89%** and the UGS technique further boosts the performance up to **72.20%**. Moreover, the STL10 dataset is the hardest for PAPs to transfer among these tasks. However, $L4A_{ugs}$ can significantly improve the cross-finetuning transferability, achieving an attack success rate of 90.05% and 98.96% for Resnet101 and ViT16 in STL10, respectively. Another intriguing finding is that ViT16 with a transformer backbone shows severe vulnerabilities to PAPs. Although performing best on legitimate samples, they bear an attack success rate of **95.30%** under the $L4A_{ugs}$ attack, closing to random outputs. These results reveal the serious security problem of the pre-training to fine-tuning paradigm and demonstrate the effectiveness of our method in such a problem setting.

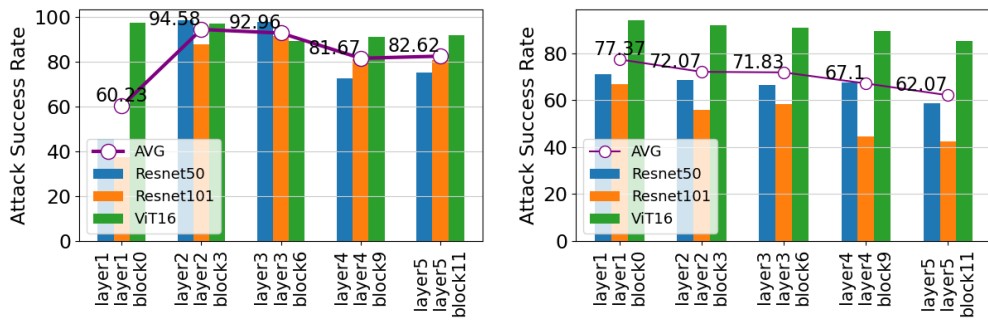

Figure 4: The attack success rates (%) of L4A$_{base}$ when using different layers. We show the results on the pre-training domain (**Left**) and fine-tuning domains (**Right**).

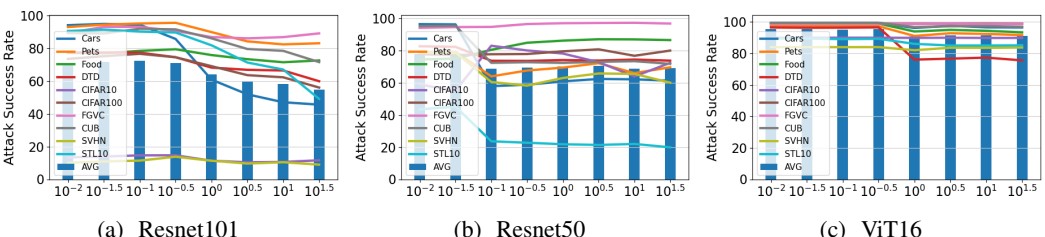

    (a)  Resnet101              (b)  Resnet50             (c)  ViT16

Figure 5: The attack success rate (%) of different hyperparameter $\lambda$ in L4A$_{ugs}$ for different models.

### 4.3 Ablation studies

#### 4.3.1 Effect of the attacking layer

We analyze the influence of attacking *different intermediate layers* of the networks on the performance of our proposed L4A$_{base}$ in the pre-training domain (ImageNet) and the fine-tuning domains (ten downstream tasks). To this end, we divide Resnet50, Resnet101, and ViT into five blocks (Details in Appendix F.1) and conduct our algorithm on them. Note that for the fine-tuning domains, the average attack success rates on the ten datasets are reported.

As shown in Fig. 4, the lower the level we choose to attack, the better our algorithm performs in the fine-tuning domains. Moreover, for the pre-training domain, attacking the middle layers of the networks results in a higher attack success rate compared to the top and bottom layers, which is also reported in existing works [29, 35, 52]. These results reveal the intrinsic property of the pre-training to fine-tuning paradigm. As the lower-level layers change less during the fine-tuning procedure, attacking the low-level layer becomes more effective when generating adversarial perturbations in the pre-training domain rather than the middle-level layers.

#### 4.3.2 Effect of uniform Gaussian sampling

We set $\boldsymbol{\mu}_l, \boldsymbol{\mu}_h, \boldsymbol{\sigma}_l, \boldsymbol{\sigma}_h$ as 0.4, 0.6, 0.05, 0.1 for all the three models as they perform best. To discuss the effect of the hyperparameter $\lambda$ in Eq. (6) fusing the base loss and the UGS loss, we select the values with a grid of 8 logarithmically spaced learning rates between $10^{-2}$ and $10^2$. The results are shown in Fig. 5. As shown in Fig. 5(a), the best attack success rate is achieved when $\lambda = 10^{-0.5}$ on Resnet101, boosting the performance by 1.52% compared to only using the Gaussian noises.

Furthermore, we study whether adopting the fixed statistics of the pre-training dataset (i.e., the mean and standard deviation of ImageNet) can help. We report the attack success rates (%) in the Table 4, where **None** refers to using no data augmentation, **ImageNet** adopts the mean and standard deviation of ImageNet

Table 4: Fixed datasets' statistics

| Model | R101 | R50 | MAE |
|---|---|---|---|
| None | 66.95 | 71.16 | 94.00 |
| ImageNet | 68.39 | 69.50 | 95.30 |
| Uniform | 72.20 | 77.80 | 95.30 |

and **Uniform** samples a pair of mean and standard deviation from the uniform distribution. As can be seen from the table, **Uniform** outperforms **None** by 4.39% on average, while **ImageNet** does not help, which means that our proposed UGS helps to avoid overfitting the pre-training domain.

### 4.4 Visualization of feature maps

We show the feature maps before and after L4A$_{\text{fuse}}$ attack in Fig. 6. The **Left** column shows the inputs of the model, while the **Middle** and the **Right** show the feature map of the pre-trained model and the fine-tuned one, respectively. The **Upper** row represents the pipeline of a clean input in Cars, and the **Lower** shows that of adversarial ones. We can see from the upper row that fine-tuning the model can make it sensitive to the defining features related to the specific domain, such as tires and lamps. However, adding an adversarial perturbation to the image can significantly lift all the activations and finally mask the useful features. Moreover, the effect of our attack could be well preserved

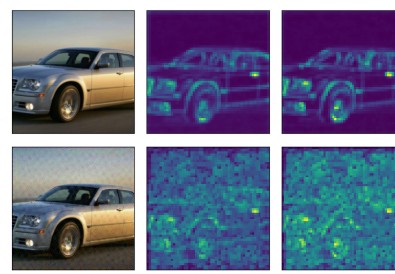

Figure 6: Visualization of feature maps.

during fine-tuning and cheat the fine-tuned model into misclassification, stressing the safety problem of pre-trained models.

### 4.5 Trade-off between the clean accuracy and robustness

We study the effect of fine-tuning epochs on the performance of our attack. To this end, we fine-tune the model until it reports the best result on the testing dataset, and then we plot the clean accuracy and the accuracy against FGSM and PAPs on Pets and STL10 in Fig. 7. The figure shows the clean accuracy and robustness of the fine-tuned model against PAPs are at odds. In Fig. 7(b), the model shows the best robustness at epoch 5 in STL10, achieving an accuracy rate of 95.38% and 55.02% on clean and adversarial samples, respec-

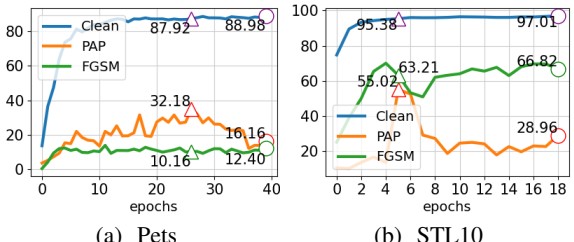

(a) Pets         (b) STL10

Figure 7: Model accuracy (%) on the Pets and STL10 datasets under clean inputs, PAP, and FGSM attack.

tively. However, the model does not converge until epoch 19. Though the process boosts the clean accuracy by **1.63%**, it suffers a significant drop in robustness, as the accuracy on adversarial samples is lowered to **28.96%**. Such findings reveal the safety problem of the pre-training to fine-tuning paradigm.

## 5 Discussion

In this section, we first introduce the gradient alignment and then use it to explain the effectiveness of our method. In particular, we show why our algorithms fall behind UAPs in the pre-training domain but have better cross-finetuning transferability when evaluated on the downstream tasks.

### 5.1 Preliminaries

**Gradient sequences.** Given a network $f_{\boldsymbol{\theta}_0}$ and a sample sequence $\{\mathbf{x}_1\, \mathbf{x}_2, \mathbf{x}_3, ..., \mathbf{x}_N\}_{\mathcal{D}}$ drawn from the dataset $\mathcal{D}$, let $\Delta\boldsymbol{\delta}_{\boldsymbol{\theta}_0,\mathcal{D}} = \{\Delta\boldsymbol{\delta}_{\boldsymbol{\theta}_0,\mathbf{x}_1}, \Delta\boldsymbol{\delta}_{\boldsymbol{\theta}_0,\mathbf{x}_2}, \Delta\boldsymbol{\delta}_{\boldsymbol{\theta}_0,\mathbf{x}_3}, ..., \Delta\boldsymbol{\delta}_{\boldsymbol{\theta}_0,\mathbf{x}_N}\}_{\mathcal{D}}$ be the sequence of gradients obtained when generating adversarial samples by the following equation:

$$\Delta\boldsymbol{\delta}_{\boldsymbol{\theta}_0,\mathbf{x}_i} = \nabla_{\boldsymbol{\delta}} L(f_{\boldsymbol{\theta}_0}, \mathbf{x}_i, \boldsymbol{\delta}_i), \text{ with } \boldsymbol{\delta}_i = P_{\infty,\boldsymbol{\epsilon}}(\boldsymbol{\delta}_{i-1} + \Delta\boldsymbol{\delta}_{\boldsymbol{\theta}_0,\mathbf{x}_{i-1}}), \tag{7}$$

where $L$ denotes the loss function of iterative attack methods like UAP, FFF, and L4A.

**Definition 1** (Gradient alignment). Given a dataset $\mathcal{D}$ and a model $f_{\boldsymbol{\theta}_0}$, the gradient alignment $\mathcal{GA}$ of an attack algorithm is defined as the expectation over the cosine similarity of $\Delta\boldsymbol{\delta}_{\boldsymbol{\theta}_0,\mathbf{x}_1}$ and $\Delta\boldsymbol{\delta}_{\boldsymbol{\theta}_0,\mathbf{x}_2}$,

which can be formulated as Eq. (8)

$$\mathcal{GA} = \mathbb{E}_{\mathbf{x}_1 \sim \mathcal{D}, \mathbf{x}_2 \sim \mathcal{D}} \left[ \frac{\Delta\boldsymbol{\delta}_{\boldsymbol{\theta}_0, \mathbf{x}_1} \cdot \Delta\boldsymbol{\delta}_{\boldsymbol{\theta}_0, \mathbf{x}_2}}{\|\Delta\boldsymbol{\delta}_{\boldsymbol{\theta}_0, \mathbf{x}_1}\|_2 \cdot \|\Delta\boldsymbol{\delta}_{\boldsymbol{\theta}_0, \mathbf{x}_2}\|_2} \right], \tag{8}$$

where $\Delta\boldsymbol{\delta}_{\boldsymbol{\theta}_0, \mathbf{x}_1}$ and $\Delta\boldsymbol{\delta}_{\boldsymbol{\theta}_0, \mathbf{x}_2}$ are two consecutive elements obtained by Eq. (7).

Then *the L4A algorithm bears a higher gradient alignment* (A strict definition and proof in a weaker form can be found in AppendixB.1). In addition, we provide the results of a simulation experiment to justify it in Table 5. We can see a negative correlation between the gradient alignment and the attack success rate on ImageNet. In contrast, a positive correlation exists between the gradient alignment and attack success rate in the fine-tuning domain.

**Effectiveness of the algorithm.** Given a pre-trained model $f_{\boldsymbol{\theta}}$ and the pre-training dataset $\mathcal{D}_p$, the generation of PAPs can be reformulated from Eq. (1) to the following equation:

$$\max_{\boldsymbol{\delta} \in \; Span\{\Delta\boldsymbol{\delta}_{\boldsymbol{\theta}, \mathcal{D}_p}\}} \mathbb{E}_{\mathbf{x} \sim \mathcal{D}_t}[F_{\boldsymbol{\theta}'}(\mathbf{x}) \neq F_{\boldsymbol{\theta}'}(\mathbf{x} + \boldsymbol{\delta})], \text{ s.t. } \|\boldsymbol{\delta}\|_p \leq \epsilon \text{ and } \mathbf{x} + \boldsymbol{\delta} \in [0, 1], \tag{9}$$

where $Span\{\Delta\boldsymbol{\delta}_{\boldsymbol{\theta}, \mathcal{D}_p}\}$ denotes the subspace spanned by the elements of $\Delta\boldsymbol{\delta}_{\boldsymbol{\theta}, \mathcal{D}_p}$. The optimal value $\boldsymbol{\delta}$ can be viewed as a linear combination of the elements in $\Delta\boldsymbol{\delta}_{\boldsymbol{\theta}, \mathcal{D}_p}$, so it is the *feasible region* of the equation. In particular, the feasible region of Eq. (9) is smaller than that of Eq. (1), which reflects that the stochastic gradient descent methods may not converge to the global maximum when it is not included in $Span\{\Delta\boldsymbol{\delta}_{\boldsymbol{\theta}, \mathcal{D}_p}\}$ and the local maximum of Eq. (9) in $Span\{\Delta\boldsymbol{\delta}_{\boldsymbol{\theta}, \mathcal{D}_p}\}$ represents the effectiveness of the algorithm in the fine-tuning domain.

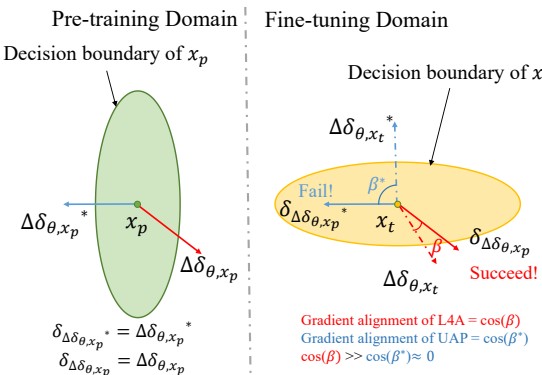

| **Resnet101** | $\mathcal{GA}$ | ImageNet | AVG |
|---|---|---|---|
| FFF$_{no}$ | 0.1221 | 38.64% | 48.55% |
| FFF$_{mean}$ | 0.0194 | 48.37% | 44.22% |
| FFF$_{one}$ | 0.1222 | 34.15% | 40.26% |
| DR | 0.0165 | 41.06% | 42.62% |
| UAP | 0.0018 | 88.14% | 43.86% |
| UAPEPGD | 0.0008 | **94.62%** | 59.39% |
| SSP | 0.0274 | 41.81% | 40.75% |
| L4A$_{base}$ | **0.6125** | 37.44% | **66.95%** |

Table 5: **Left:** Gradient alignments; **Middle:** Attack success rates on the pre-training dataset; **Right:** Average attack success rates on downstream tasks. See more details in AppendixB.2

Figure 8: Illustration

## 5.2 Explanation

We aim to explain why the effectiveness of our algorithm is better in the fine-tuning domain and worse in the pre-training domain, as seen from Table 5. Let $\Delta\boldsymbol{\delta}_{\boldsymbol{\theta}, \mathcal{D}_p}$ and $\Delta\boldsymbol{\delta}_{\boldsymbol{\theta}, \mathcal{D}_p}^*$ be the feasible zones of L4A and UAP obtained by feeding instances from $D_p$ into the pre-trained model $f_{\boldsymbol{\theta}}$, respectively. Similarly, we can define $\Delta\boldsymbol{\delta}_{\boldsymbol{\theta}, \mathcal{D}_t}$ and $\Delta\boldsymbol{\delta}_{\boldsymbol{\theta}, \mathcal{D}_t}^*$. Meanwhile, denote $\boldsymbol{\delta}_{\Delta\boldsymbol{\delta}, \mathcal{D}_p}$ and $\boldsymbol{\delta}_{\Delta\boldsymbol{\delta}, \mathcal{D}_p}^*$ as the maxima in the pre-training domain obtained by L4A and UAP respectively. An illustration is shown in Fig. 8, supposing there is only one step in the iterative method.

**Pre-training domain:** Because the gradients of UAP obtained by Eq. (2) represent the directions to the closest points on the decision boundary in the pre-training domain. Thus, limiting the feasible zone to $\Delta\boldsymbol{\delta}_{\boldsymbol{\theta}, \mathcal{D}_p}^*$ does little harm to the performance when evaluated in the pre-training domain. Meanwhile, according to the optimal objective, L4A finds the next best directions which are worse than those of UAPs. Thus, $\boldsymbol{\delta}_{\Delta\boldsymbol{\delta}, \mathcal{D}_p}^*$ performs better than $\boldsymbol{\delta}_{\Delta\boldsymbol{\delta}, \mathcal{D}_p}$ in the pre-training domain.

**Fine-tuning domain:** According to the fact that UAP bears a low gradient alignment near to 0, the subspace spanned by the tensors in $\Delta\boldsymbol{\delta}_{\boldsymbol{\Delta}\boldsymbol{\theta}, \mathcal{D}_p}^*$ is almost orthogonal to that spanned by the tensors in $\Delta\boldsymbol{\delta}_{\boldsymbol{\Delta}\boldsymbol{\theta}, \mathcal{D}_t}^*$ which represent the best directions that send the sample to the decision boundary in the fine-tuning domain. Thus, limiting the feasible zone of Eq. (1) to $\Delta\boldsymbol{\delta}_{\boldsymbol{\theta}, \mathcal{D}_p}^*$ suffers a great drop in ASR when evaluated on the downstream tasks in Eq. (9). However, as shown in Table 5, our

algorithm can achieve a gradient alignment of up to 0.6125, which means that there is considerable overlap in the next best feasible region of $\mathcal{D}_p$ in the pre-training domain obtained by L4A and that of $\mathcal{D}_f$ in the fine-tuning domain. Thus the performance of the best solution in $\Delta\boldsymbol{\delta}_{\boldsymbol{\theta},\mathcal{D}_p}$ is close to that of $\Delta\boldsymbol{\delta}_{\boldsymbol{\theta},\mathcal{D}_f}$, which represents the next best solution in the fine-tuning domain. Finally we have $\boldsymbol{\delta}_{\Delta\boldsymbol{\delta}_{\boldsymbol{\theta},\mathcal{D}_p}}$ performs better than $\boldsymbol{\delta}^*_{\Delta\boldsymbol{\delta}_{\boldsymbol{\theta},\mathcal{D}_p}}$ in the fine-tuning domain.

In conclusion, *the high gradient alignment guarantees high cross-finetuning transferability*.

# 6 Societal impact

A potential negative societal impact of L4A is that malicious adversaries could use it to cause security/safety issues in real-world applications. As more people focus on the pre-trained models because of their excellent performance, fine-tuning pre-trained models provided by the cloud server becomes a panacea for deep learning practitioners. In such settings, PAPs become a significant security flaw–as one can easily access the prototype pre-trained models and perform attacking algorithms on them. Our work appeals to big companies to delve further into the safety problem related to the vulnerability of pre-trained models.

# 7 Conclusion

In this paper, we address the safety problem of pre-trained models. In particular, an attacker can use them to generate so-called pre-trained adversarial perturbations, achieving a high success rate on the fine-tuned models without knowing the victim model and the specific downstream tasks. Considering the inner qualities of the pre-training to fine-tuning paradigm, we propose a novel algorithm, L4A, which performs well in such problem settings. A limitation of L4A is that it performs worse than UAPs in the pre-training domain; we hope some upcoming work can fill the gap. Furthermore, L4A only utilizes the information in the pre-training domain. When the attacker obtains some information about the downstream tasks, like several unlabeled instances in the fine-tuning domain, he may be able to enhance PAPs using the knowledge and further exacerbate the situation, which we leave to future work. Thus, we hope our work can draw attention to the safety problem of pre-trained models to guarantee security.

# Acknowledgement

This work was supported by the National Key Research and Development Program of China (2020AAA0106000, 2020AAA0104304, 2020AAA0106302), NSFC Projects (Nos. 62061136001, 62076145, 62076147, U19B2034, U1811461, U19A2081, 61972224), Beijing NSF Project (No. JQ19016), BNRist (BNR2022RC01006), Tsinghua Institute for Guo Qiang, and the High Performance Computing Center, Tsinghua University. Y. Dong was also supported by the China National Postdoctoral Program for Innovative Talents and Shuimu Tsinghua Scholar Program.

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
