# A  Additional experiments

## A.1  Additional experiments on other pre-trained models

In this section, we report the results on CLIP and MOCO in Table 1 and Table 2, respectively. Note that the first seven columns of validation datasets are fine-grained, while the next three are coarse-grained ones. We mark the best results for each dataset in **bold**, and the best baseline in **blue**. We highlight the results of L4A$_{ugs}$ in red to emphasize that the Low-Lever Layer Lifting attack equipped with Uniform Gaussian Sampling shows great *cross-finetuning transferability* and performs best.

These tables show that our proposed methods outperform all the baselines by a large margin. For example, as can be seen from Table 1, if the target model is Resnet50 pre-trained by MOCO, the best competitor UAP achieves an average attack success rate of 54.34%, while the L4A$_{fuse}$ can lift it up to **54.72%** and the UGS technique further boosts the performance up to **59.72%**.

Table 1: The attack success rate(%) of various methods we study against **Resnet50** pretrained by **MOCO**. Note that C10 stands for CIFAR10, and C100 stands for CIFAR100.

| ASR | Cars | Pets | Food | DTD | FGVC | CUB | SVHN | C10 | C100 | STL10 | AVG |
|---|---|---|---|---|---|---|---|---|---|---|---|
| FFF$_{no}$ | 30.72 | 25.59 | 60.03 | 48.51 | 84.97 | 62.82 | 4.92 | 17.22 | 55.37 | 12.84 | 40.30 |
| FFF$_{mean}$ | 39.04 | 32.49 | 68.94 | 52.93 | 85.57 | **68.93** | 8.02 | 22.23 | 67.53 | 14.26 | 45.99 |
| FFF$_{one}$ | 31.82 | 27.26 | 53.06 | 51.38 | 77.92 | 54.42 | 5.09 | 17.43 | 57.26 | 10.23 | 38.59 |
| DR | 44.30 | 39.63 | 51.11 | 53.51 | 81.43 | 55.85 | 5.01 | 30.24 | 74.25 | 13.91 | 44.92 |
| SSP | 33.75 | 36.44 | 75.32 | 60.00 | 80.32 | 68.73 | 5.90 | 25.42 | 69.27 | 29.03 | 48.42 |
| ASV | 40.01 | 32.27 | 60.63 | 47.50 | 82.87 | 62.77 | **41.71** | 11.15 | 50.30 | 9.05 | 43.83 |
| UAP | 61.00 | 52.44 | 77.00 | 60.75 | 83.79 | 68.54 | 5.75 | 28.02 | 68.22 | 37.94 | 54.34 |
| UAPEPGD | 45.55 | 38.05 | 70.12 | 54.79 | 69.04 | 57.25 | 3.82 | 10.47 | 48.38 | 21.44 | 41.89 |
| L4A$_{base}$ | 44.10 | 51.86 | 77.44 | 62.61 | 81.49 | 61.30 | 5.65 | 45.70 | 81.88 | 27.33 | 53.94 |
| L4A$_{fuse}$ | 44.25 | 54.02 | 78.09 | 63.19 | 82.90 | 63.26 | 5.13 | **46.71** | 81.66 | 27.95 | 54.72 |
| L4A$_{ugs}$ | **61.22** | **58.11** | **86.52** | **67.71** | **88.96** | 65.57 | 5.08 | 39.23 | **83.12** | **41.93** | **59.74** |

Table 2: The attack success rate(%) of various methods we study against **Resnet50** pretrained by **CLIP**. Note that C10 stands for CIFAR10, and C100 stands for CIFAR100.

| ASR | Cars | Pets | Food | DTD | FGVC | CUB | SVHN | C10 | C100 | STL10 | AVG |
|---|---|---|---|---|---|---|---|---|---|---|---|
| FFF$_{no}$ | 89.96 | 90.41 | 94.57 | 82.45 | 87.81 | **99.08** | 98.50 | 78.20 | 80.41 | 89.19 | 89.06 |
| FFF$_{mean}$ | 92.86 | 89.78 | 93.28 | 81.49 | 89.30 | 99.00 | 99.10 | 75.35 | 80.41 | 89.71 | 89.03 |
| FFF$_{one}$ | 89.21 | 86.73 | 93.47 | 80.32 | 85.63 | 99.03 | 98.38 | 71.78 | 80.41 | 84.89 | 86.98 |
| DR | 71.84 | 52.74 | 77.07 | 70.32 | 86.69 | 99.01 | 94.69 | 61.74 | 80.41 | 84.06 | 77.86 |
| SSP | 85.61 | 84.98 | 83.23 | 78.19 | 90.00 | 98.98 | 98.50 | 75.80 | 80.41 | 87.90 | 86.36 |
| ASV | 91.04 | 90.80 | 93.98 | 81.82 | 88.96 | 98.02 | 97.68 | 78.75 | 79.41 | 87.34 | 88.78 |
| UAP | 75.84 | 65.84 | 87.12 | 73.40 | 88.08 | 99.00 | 96.52 | 57.18 | 80.41 | 87.03 | 81.04 |
| UAPEPGD | 64.15 | 52.14 | 60.35 | 67.23 | 89.99 | 98.59 | 95.14 | 47.05 | 80.41 | 80.86 | 73.59 |
| L4A$_{base}$ | 79.83 | 66.61 | 79.74 | 70.53 | 85.96 | 99.03 | 98.04 | 64.00 | 84.01 | 82.26 | 81.00 |
| L4A$_{fuse}$ | 96.57 | 96.27 | 98.39 | 86.01 | 88.11 | 99.00 | 98.41 | 80.41 | **80.95** | **90.01** | 91.41 |
| L4A$_{ugs}$ | **97.12** | **97.19** | **98.67** | **91.12** | **91.56** | 99.02 | **99.10** | **96.55** | 80.41 | 85.36 | **93.61** |

## A.2  PAPs against adversarial fine-tuned models

In our paper, we only conduct experiments on standard fine-tuning. We followed the training process introduced by [5], which adopts a distillation term to preserve high-quality features of the pretrained model to boost model performance from a view of information theory. We do some additional experiments on **Resnet50** pretrained by **SimCLRv2**. And the results are as follows.

Table 3: The attack success rate(%) of different methods against adversarial fine-tuned models. Note that C10 stands for CIFAR10 and C100 stands for CIFAR100.

| ASR | Cars | Pets | Food | DTD | FGVC | CUB | SVHN | C10 | C100 | STL10 | AVG |
|---|---|---|---|---|---|---|---|---|---|---|---|
| FFF$_{no}$ | 14.69 | 26.08 | 50.86 | 48.40 | 27.68 | 31.74 | 34.35 | 36.78 | 10.30 | 16.56 | 29.74 |
| FFF$_{mean}$ | **14.72** | 26.30 | 50.36 | 48.14 | 27.70 | 31.90 | 33.93 | 36.74 | 9.97 | 16.61 | 29.64 |
| FFF$_{one}$ | 14.65 | 26.53 | 53.08 | 48.52 | 27.06 | 31.49 | **34.02** | 37.35 | 10.99 | 16.73 | 30.04 |
| DR | 14.33 | 26.96 | 50.84 | 47.13 | 27.39 | 32.41 | 33.81 | 37.39 | 10.85 | 16.30 | 29.74 |
| SSP | 14.31 | 26.08 | 50.01 | 48.08 | **27.51** | 32.10 | 33.14 | 37.14 | 11.20 | 16.34 | 29.59 |
| ASV | 11.95 | 16.35 | 25.37 | 37.23 | 24.58 | 26.93 | 32.22 | 31.39 | 6.32 | 7.15 | 21.95 |
| UAP | 14.54 | 25.67 | 47.92 | 48.19 | 27.34 | 32.60 | 33.60 | 36.71 | 10.94 | 16.26 | 29.38 |
| UAPEPGD | 14.40 | 24.20 | 46.82 | 47.18 | 27.32 | 32.53 | 34.02 | 35.29 | 10.38 | 15.51 | 28.77 |
| L4A$_{base}$ | 14.48 | 26.76 | **55.66** | 50.69 | 26.22 | 32.14 | 33.45 | 37.33 | 11.10 | 16.75 | 30.46 |
| L4Afuse | 14.64 | **28.07** | 55.59 | **50.74** | 26.96 | **32.49** | 33.57 | **37.66** | **11.31** | **16.94** | **30.80** |
| L4A$_{ugs}$ | 14.35 | 25.84 | 53.79 | 50.16 | 26.80 | 32.02 | 33.54 | 36.52 | 10.81 | 16.56 | 30.04 |

As seen from the table, all the methods suffer degenerated performance against adversarial fine-tuning. However, L4A still performs best among these competitors. For example, considering the DTD dataset, the best baseline FFF$_{one}$ achieves an attack success rate of 48.52%, while the ASR of the villain L4A$_{base}$ is up to 50.69%, and the fusing loss further boosts the performance to 50.74%.

## A.3    PAPs against adversarial pretrained models

To test PAPs against adversarial pretrained models, we follow the method proposed by [6], which uses adversarial views to boost robustness. We first train a robust Resnet50 using enhanced Sim-CLRv2. Then, we generate PAPs using that model and test them on both the adversarial fine-tuned and standard fine-tuned models on downstream tasks. In Table 4, we report the results on standard finetuned models.

Table 4: The attack success rate(%) of different methods against adversarial-pretrained-standard-finetuned models. Note that C10 stands for CIFAR10, and C100 stands for CIFAR100.

| ASR | Cars | Pets | Food | DTD | FGVC | CUB | SVHN | C10 | C100 | STL10 | AVG |
|---|---|---|---|---|---|---|---|---|---|---|---|
| FFF$_{no}$ | 45.34 | 23.33 | 53.16 | 44.95 | 67.63 | 45.89 | 81.12 | 60.78 | 95.38 | 18.48 | 53.61 |
| FFF$_{mean}$ | 50.81 | 27.86 | 60.99 | 46.97 | 74.56 | 52.69 | 81.27 | 71.33 | 99.00 | 28.97 | 59.45 |
| FFF$_{one}$ | 44.37 | 25.21 | 47.12 | 46.65 | 68.35 | 43.89 | 81.50 | 63.36 | 97.78 | 15.10 | 53.33 |
| DR | 48.81 | 34.15 | 61.32 | 48.30 | 79.98 | 51.36 | 76.29 | 63.17 | 80.29 | 13.22 | 55.69 |
| SSP | 44.67 | **34.23** | 44.84 | 48.72 | 69.79 | 45.32 | 83.32 | 81.44 | 97.95 | 20.95 | 54.21 |
| ASV | 47.94 | 25.32 | 56.66 | 45.80 | 65.94 | 41.14 | 78.52 | 61.25 | 96.17 | 12.16 | 53.09 |
| UAP | 35.03 | 32.65 | 38.39 | 46.44 | 67.12 | 42.22 | 78.72 | 67.66 | 93.81 | **36.41** | 53.84 |
| UAPEPGD | 23.20 | 17.85 | 25.20 | 39.10 | 53.47 | 31.19 | 66.41 | 13.63 | 66.98 | 6.33 | 34.33 |
| L4A$_{base}$ | 55.93 | 26.41 | 67.82 | 52.55 | 79.60 | 56.64 | 84.07 | **84.52** | 99.00 | 28.16 | 63.47 |
| L4A$_{fuse}$ | 67.08 | 25.94 | 67.95 | **54.04** | 82.20 | 61.41 | 84.05 | 83.09 | **99.00** | 33.71 | 65.85 |
| L4A$_{ugs}$ | **82.46** | 25.91 | **74.37** | 51.65 | **88.93** | **72.14** | **84.08** | 78.66 | 98.98 | 31.97 | **68.91** |

The above table shows that adversarial-pretrained models show little robustness after standard fine-tuning, which is also reported in [2, 9]. In such settings, L4A still performs best among these competitors: the best baseline FFF$_{mean}$ achieves an average attack success rate of 59.45%, while the ASR of the villain L4A$_{base}$ is up to 63.47%, and the Uniform Gaussian sampling further boosts the performance to 68.91%. Another interesting finding is that low-level-based methods, such as FFF, DR, SSP and L4A, perform better than high-level-based ones like UAPEPGD, which uses classification scores. This further supports our findings in Fig 2 and our motivation to use low-level layers.

In Table 5, we report the results on adversarial-finetuned models. Note that we adopt the adversarial-finetuning method in [5].

Table 5: The attack success rate(%) of different methods against adversarial-pretrained-adversarial-fine-tuned models. Note that C10 stands for CIFAR10, and C100 stands for CIFAR100.

| ASR | Cars | Pets | Food | DTD | FGVC | CUB | SVHN | C10 | C100 | STL10 | AVG |
|---|---|---|---|---|---|---|---|---|---|---|---|
| $FFF_{no}$ | 14.82 | 27.13 | 47.84 | 49.19 | 37.61 | 38.82 | 9.24 | 21.27 | 39.32 | 16.60 | 30.18 |
| $FFF_{mean}$ | 14.88 | 27.09 | 48.82 | 49.40 | 38.01 | 38.80 | 9.09 | 20.45 | 39.32 | 16.47 | 30.23 |
| $FFF_{one}$ | 14.95 | 27.25 | 47.08 | 49.40 | 37.80 | 39.01 | 9.02 | 21.09 | 39.31 | 16.31 | 30.12 |
| DR | 15.18 | 28.59 | 48.51 | 48.30 | 38.46 | 38.73 | 8.64 | 20.73 | **39.61** | 16.86 | 30.36 |
| SSP | 15.20 | 28.67 | 39.93 | 46.38 | **38.88** | 37.42 | 7.35 | 21.89 | 39.49 | 15.11 | 29.03 |
| ASV | 15.56 | 27.77 | 49.17 | 49.15 | 38.58 | 37.83 | 10.16 | 18.82 | 34.71 | 13.44 | 29.52 |
| UAP | 15.53 | 27.77 | 48.23 | 46.96 | 38.55 | 37.59 | 7.49 | 15.64 | 35.30 | 14.43 | 28.75 |
| UAPEPGD | 15.00 | 28.24 | 41.39 | 46.49 | 37.98 | 37.04 | 6.69 | 11.75 | 35.02 | 12.44 | 27.20 |
| $L4A_{base}$ | 15.64 | 29.59 | 49.18 | 51.65 | 38.01 | 39.83 | 11.24 | 24.43 | 38.39 | 16.96 | 31.49 |
| $L4A_{fuse}$ | **15.90** | **30.88** | **49.83** | 49.89 | 38.01 | **40.05** | 10.69 | 24.25 | 37.20 | 17.10 | 31.38 |
| $L4A_{ugs}$ | 15.69 | 30.25 | 49.59 | **52.02** | 37.68 | 39.87 | **11.25** | **24.52** | 38.06 | **17.11** | **31.60** |

From Table 5, we can see that adversarial-pretrained-adversarial-finetuned models show much robustness after fine-tuning, which is consistent with Appendix A.2 and the finding in [2, 9] that adversarial fine-tuning contributes to the final robustness more than adversarial pre-training. Although all the methods degenerate a lot, L4A is still among the best ones.

### A.4 PAPs in other vision tasks

We conduct experiments on semantic segmentation and object detection tasks in this subsection to evaluate our methods. For object detection, we adopt the off-the-shelf Resnet50 model provided by MMDetection repo, which is pre-trained by the method of MOCOv2 on ImageNet and then fine-tuned on the COCO object detection task. The results of different methods are in Table 6

Table 6: Objection detection finetuned on COCO. Evaluation is on COCO val2017, and results are reported in the metrics of mAP, $mAP_{50}$, and $mAP_{75}$. We mark the best ones for each metric in **bold**. Note that EPGD stands for the UAPEPGD method.

| Methods | $FFF_{no}$ | $FFF_{mean}$ | $FFF_{one}$ | STD | SSP | ASV | UAP | EPGD | $L4A_{base}$ | $L4A_{fuse}$ | $L4A_{ugs}$ |
|---|---|---|---|---|---|---|---|---|---|---|---|
| mAP | 30.7 | 30.0 | 30.8 | 31.6 | 31.0 | 29.8 | 30.2 | 34.2 | 29.8 | 29.3 | **26.5** |
| $mAP_{50}$ | 48.5 | 47.6 | 48.6 | 49.6 | 48.6 | 46.9 | 47.8 | 53.1 | 46.9 | 46.2 | **42.5** |
| $mAP_{75}$ | 32.9 | 32.1 | 33.0 | 34.2 | 33.4 | 31.9 | 32.5 | 37.4 | 32.0 | 31.6 | **28.2** |

The table shows that our proposed methods outperform all the baselines by a large margin. For example, the best competitor ASV achieves a $mAP_{50}$ of 46.9%, while the UGS technique can degenerate it to **42.5%**, showing its effectiveness.

As for segmentation, we use the ViT-base model provided by MMSegmentatation, which is pre-trained by MAE on ImageNet and then finetuned on the ADE20k dataset. The results are as follows:

Table 7: Segmentation finetuned on ADE20k. Results are reported in the metric of mIoU. Note that EPGD stands for UAPEPGD method.

| methods | $FFF_{no}$ | $FFF_{mean}$ | $FFF_{one}$ | STD | SSP | ASV | UAP | EPGD | $L4A_{base}$ | $L4A_{fuse}$ | $L4A_{ugs}$ |
|---|---|---|---|---|---|---|---|---|---|---|---|
| mIoU | 40.63 | 40.79 | 40.86 | 42.67 | 41.99 | 41.84 | 41.89 | 41.07 | **39.38** | 39.59 | 39.44 |

From the table, we can see that our methods generalize well to the segmentation task. While $FFF_{no}$ achieve a mIoU of 40.63%, that of $L4A_{no}$ is 39.38%. All the experiments above show the great cross-task transfer-ability of our methods.

## B  Gradient alignment

### B.1  Gradient alignment: Proof

In this section, we formulate the definition of the gradient alignment and give a brief proof.

### B.1.1 Preliminaries

Given a convolutional layer $Conv$ with kernel size = $ks$, stride = 1, bias = 0, an input image $\mathbf{im} \in \mathcal{R}^{in \times in}$, then the output $ReLu\big[Conv(\mathbf{im}, kernel)\big] \in \mathcal{R}^{(in-ks+1) \times (in-ks+1)}$. Note that $ks << in$.

According to the methods for calculating convolution in computers, the input image $\mathbf{x}$ will be flatten into a vector $\mathbf{x} \in \mathcal{R}^n$, where $n = in \times in$ and the weights of the convolution layer can be reshaped into a matrix $\mathbf{W} \in \mathcal{R}^{m \times n}$, where $m = (in - ks + 1) \times (in - ks + 1)$. Then we have the output $\mathbf{y} = ReLu\big[\mathbf{Wx}\big]$.

Denote $\mathbf{w}_i$ as the i-th row of the matrix $\mathbf{W}$. Let the elements of the kernel and $\mathbf{x}$ subject to the standard normal distribution independently.

**Lemma 1.** $\mathbb{E}\big[\mathbf{w}_i \mathbf{w}_j^T\big] = ks^2 \delta_{i,j}$.

*Proof.* When $i = j$, we have $\mathbf{w}_i \mathbf{w}_i^T \sim \mathcal{X}^2(ks^2)$. Thus $\mathbb{E}\big[\mathbf{w}_i \mathbf{w}_i^T\big] = ks^2$.
When $i \neq j$, due to the arrangement of the none-zero elements in the matrix $\mathcal{W}$, we have $\mathbf{w}_i \mathbf{w}_j^T = \sum_{k=1}^{N} x_{k_1} x_{k_2}$, where $x_{k_1}, x_{k_2} \sim N(0,1)$ independently and $0 \leq N \leq ks^2$. Thus $\mathbb{E}\big[\mathbf{w}_i \mathbf{w}_j^T\big] = \sum_{k=1}^{N} \mathbb{E}\big[x_{k_1} x_{k_2}\big] = \sum_{k=1}^{N} \mathbb{E}\big[x_{k_1}\big] \mathbb{E}\big[x_{k_2}\big] = 0$ □

**Lemma 2.** $\mathcal{P}\big(\mathbf{w}_i \mathbf{w}_j^T = 0\big) = \dfrac{\binom{n-ks^2}{ks^2}}{\binom{n}{ks^2}}$.

*Proof.* There are only $ks^2$ non-zero elements in $\mathbf{w}_i$. Then we have $\binom{n-ks^2}{ks^2}$ ways to choose $ks^2$ zero elements from $\mathbf{w}_j$, making the sum of the product zero. Meanwhile, we have $\binom{n}{ks^2}$ ways to choose $ks^2$ elements from $\mathbf{w}_j$. Finally the probability is the ratio of the two values. □

**Assumption 1.** $\mathbf{w}_i \mathbf{w}_j^T = 0$, for $i \neq j$.

According to Lemma 1, $\mathbb{E}\big[\mathbf{w}_i \mathbf{w}_j^T\big] = 0$ for $i \neq j$. According to Lemma 2, $\lim\limits_{\frac{ks^2}{n} \to 0} \mathcal{P}\big(\mathbf{w}_i \mathbf{w}_j^T = 0\big) = 1$.

**Assumption 2.** The elements of $\mathbf{x_1}$, $\mathbf{x_2}$ and the kernel subject to the standard normal distribution independently.

### B.1.2 Proof

Let $\mathbf{x}_1$ and $\mathbf{x}_2 \in \mathcal{R}^n$ be two flattened vectors and let $\mathbf{W}$ be the weight matrix of the first convolution layer.

For the first iteration of the L4A algorithm, the output of the first convolution layer $\mathbf{y}_1 = ReLu\big(\mathbf{Wx}_1\big)$. Then the gradient of the loss function in the first step is:

$$
\begin{aligned}
\frac{\partial L}{\partial \mathbf{x}_1} &= \frac{1}{m} \frac{\partial \mathbf{y}_1^T \mathbf{y}_1}{\partial \mathbf{x}_1} = \frac{1}{m} \sum_{i=1}^{m} \frac{\partial ReLu^2(\mathbf{w}_i \mathbf{x}_1)}{\mathbf{x}_1} \mathbf{w}_i \\
&= \frac{2}{m} \sum_{i=1}^{m} ReLu(\mathbf{w}_i \mathbf{x}_1) U(\mathbf{w}_i \mathbf{x}_1) \mathbf{w}_i = \frac{2}{m} \sum_{i=1}^{m} ReLu(\mathbf{w}_i \mathbf{x}_1) \mathbf{w}_i
\end{aligned}
\tag{1}
$$

where $U(\cdot)$ denotes the step function.

Considering the update of $\boldsymbol{\delta}$, we have the output in the second step $\mathbf{y}_2 = ReLu\big[\mathbf{W}\big(\mathbf{x}_1 + \boldsymbol{\alpha} \frac{\partial L}{\partial \mathbf{x}_1}\big)\big]$ where $\alpha$ denotes the step size. Then the gradient of the loss function in the second step is:

$$\frac{\partial L}{\partial \mathbf{x}_2} = \frac{1}{m}\frac{\partial \mathbf{y}_2^T \mathbf{y}_2}{\partial \mathbf{x}_2} = \frac{1}{m}\sum_{i=1}^{m}\frac{\partial ReLu^2\big(\mathbf{w}_i\mathbf{x}_2 + \mathbf{w}_i\frac{2\alpha}{m}\sum_{j=1}^{m}ReLu\big(\mathbf{w}_j\mathbf{x}_1\big)\mathbf{w}_j^T\big)}{\partial \mathbf{x}_2}$$

$$= \frac{1}{m}\sum_{i=1}^{m}\frac{\partial ReLu^2\big(\mathbf{w}_i\mathbf{x}_2 + \frac{2\alpha}{m}ReLu\big(\mathbf{w}_i\mathbf{x}\big)\mathbf{w}_i\mathbf{w}_i^T\big)}{\partial \mathbf{x}_2} \qquad (2)$$

$$= \frac{2}{m}\sum_{i=1}^{m}ReLu\big(\mathbf{w}_i\mathbf{x}_2 + \frac{2\alpha}{m}ReLu\big(\mathbf{w}_i\mathbf{x}\big)\mathbf{w}_i\mathbf{w}_i^T\big)\mathbf{w}_i$$

Ignoring the update of $\boldsymbol{\delta}$, we have the output of the convolution layer in the second step $\mathbf{y}_2^* = ReLu\big(\mathbf{W}\mathbf{x}_2\big)$

Then the gradient in the second step can be formulated as:

$$\frac{\partial L^*}{\partial \mathbf{x}_2} = \frac{1}{m}\frac{\partial \mathbf{y}_2^{*T}\mathbf{y}_2^*}{\partial \mathbf{x}_2} = \frac{1}{m}\sum_{i=1}^{m}\frac{\partial ReLu^2\big(\mathbf{w}_i\mathbf{x}_2\big)}{\mathbf{x}_2}\mathbf{w}_i$$

$$= 2\frac{1}{m}\sum_{i=1}^{m}ReLu\big(\mathbf{w}_i\mathbf{x}_2\big)U\big(\mathbf{w}_i\mathbf{x}_2\big)\mathbf{w}_i = 2\frac{1}{m}\sum_{i=1}^{m}ReLu\big(\mathbf{w}_i\mathbf{x}_2\big)\mathbf{w}_i \qquad (3)$$

**Definition 1** (Gradient alignment). $\mathcal{GA} = \mathbb{E}_{\mathbf{x}_1,\mathbf{x}_2}\big[\frac{\partial L}{\partial \mathbf{x}_2}\frac{\partial L}{\partial \mathbf{x}_1}^T\big]$

Here the gradient alignment measures the similarity between steps of different iterations.

**Definition 2** (Pseudo-gradient alignment). $\mathcal{PGA} = \mathbb{E}_{\mathbf{x}_1,\mathbf{x}_2}\big[\frac{\partial L^*}{\partial \mathbf{x}_2}\frac{\partial L}{\partial \mathbf{x}_1}^T\big]$

Here the pseudo-gradient alignment shows the similarity when ignoring the update and provides the reference value for easy comparison.

**Theorem 1.** The $\mathcal{GA}$ of L4A is never smaller than the $\mathcal{PGA}$.

*Proof.*

$$\mathbb{E}\big(\frac{\partial L}{\partial \mathbf{x}_2}\cdot\frac{\partial L}{\partial \mathbf{x}_1}^T\big) - \mathbb{E}\big(\frac{\partial L^*}{\partial \mathbf{x}_2}\cdot\frac{\partial L}{\partial \mathbf{x}_1}^T\big)$$

$$= \frac{4}{m^2}\mathbb{E}\bigg[\sum_{i=1}^{m}ReLu\big(\mathbf{w}_i\mathbf{x}_2 + \frac{2\alpha}{m}ReLu\big(\mathbf{w}_i\mathbf{x}_1\big)\mathbf{w}_i\mathbf{w}_i^T\big)\mathbf{w}_i\sum_{j=1}^{m}ReLu\big(\mathbf{w}_j\mathbf{x}_1\big)\mathbf{w}_j^T\bigg]$$

$$- \frac{4}{m^2}\mathbb{E}\bigg[\sum_{i=1}^{m}ReLu\big(\mathbf{w}_i\mathbf{x}_2\big)\mathbf{w}_i\sum_{j=1}^{m}ReLu\big(\mathbf{w}_j\mathbf{x}_1\big)\mathbf{w}_j^T\bigg]$$

$$= \frac{4}{m^2}\mathbb{E}\bigg[\sum_{i=1}^{m}ReLu\big(\mathbf{w}_i\mathbf{x}_2 + \frac{2\alpha}{m}ReLu\big(\mathbf{w}_i\mathbf{x}_1\big)\mathbf{w}_i\mathbf{w}_i^T\big)\mathbf{w}_iReLu\big(\mathbf{w}_i\mathbf{x}_1\big)\mathbf{w}_i^T\bigg]$$

$$- \frac{4}{m^2}\mathbb{E}\bigg[\sum_{i=1}^{m}ReLu\big(\mathbf{w}_i\mathbf{x}_2\big)\mathbf{w}_iReLu\big(\mathbf{w}_i\mathbf{x}_1\big)\mathbf{w}_i^T\bigg]$$

$$= \frac{4}{m^2}\sum_{i=1}^{m}\mathbb{E}\bigg[\bigg(ReLu\big(\mathbf{w}_i\mathbf{x}_2 + \frac{2\alpha}{m}ReLu\big(\mathbf{w}_i\mathbf{x}_2\big)\mathbf{w}_i\mathbf{w}_i^T\big) - ReLu\big(\mathbf{w}_i\mathbf{x}_2\big)\bigg)ReLu\big(\mathbf{w}_i\mathbf{x}_1\big)\mathbf{w}_i\mathbf{w}_i^T\bigg] \geq 0$$

Note that $ReLu\big(\mathbf{w}_i\mathbf{x}_2\big)\mathbf{w}_i\mathbf{w}_i^T \geq 0$, thus $ReLu\big(\mathbf{w}_i\mathbf{x}_2 + \frac{2\alpha}{m}ReLu\big(\mathbf{w}_i\mathbf{x}_2\big)\mathbf{w}_i\mathbf{w}_i^T\big) - ReLu\big(\mathbf{w}_i\mathbf{x}_2\big) \geq 0$ $\qquad\square$

## B.2 Gradient alignment: Simulation

In this subsection, we provide details about the simulation evaluating the gradient alignment of different algorithms. First, we run the targeted algorithm 256 times and record the gradients obtained in the algorithm. Then we compute the cosine similarity matrix of the 256 gradients and exclude diagonal elements. Finally, we refer to the average over the similarity matrix as the gradient alignment of the method. Here we provide additional simulation results on Resnet50 and ViT16 in Table 8 and Table 9 respectively. Note that we report the attack success rates (%).

| **Resnet50** | $\mathcal{GA}$ | ImageNet | AVG |
|---|---|---|---|
| FFF$_{\text{mean}}$ | 0.0158 | 45.98 | 52.10 |
| DR | 0.0449 | 45.46 | 48.26 |
| UAP | 0.0020 | 95.34 | 55.16 |
| UAPEPGD | 0.0010 | 93.67 | 69.28 |
| SSP | 0.0449 | 44.28 | 53.64 |
| L4A$_{\text{base}}$ | 0.5489 | 45.54 | 71.16 |

Table 8: Gradient alignment on Resnet50

| **ViT16** | GA | ImageNet | AVG |
|---|---|---|---|
| FFF$_{\text{mean}}$ | 0.1005 | 99.88 | 71.06 |
| DR | 0.0861 | 56.06 | 27.02 |
| UAP | 0.0504 | 98.46 | 55.16 |
| UAPEPGD | 0.0049 | 97.66 | 66.95 |
| SSP | 0.1279 | 80.63 | 53.40 |
| L4A$_{\text{base}}$ | 0.1386 | 94.15 | 94.00 |

Table 9: Gradient alignment on ViT16

# C  Ablation studies

## C.1  Effect of fusing the knowledge of different layers

Here we discuss the effect of the scale factor $\lambda$ to fuse the knowledge from different layers, and the results are shown in Fig 1. For Resnet50, in Fig. (b), setting $\lambda$ as $10^{0.5}$ can boost the performance by 1.5%.

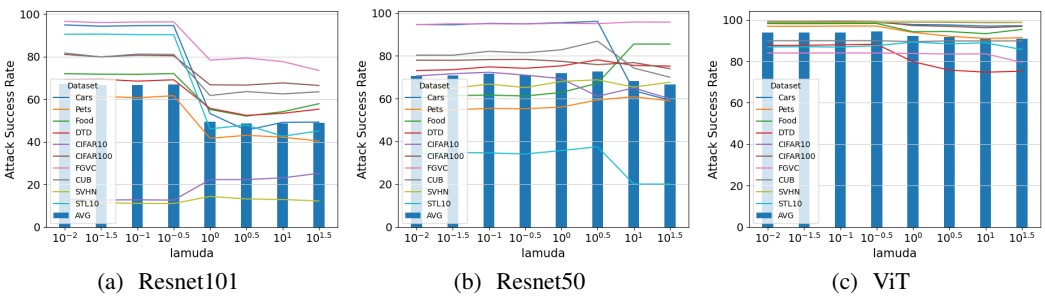

(a)  Resnet101                 (b)  Resnet50                 (c)  ViT

Figure 1: The effect of the scale factor in L4A$_{\text{fuse}}$

## C.2  Effect of using high-level loss

To study the effect of utilizing the high-level features, we choose Resnet50 pre-trained by SimCLRv2 as the target. In previous experiments, we found that the $L4A_{\text{fuse}}$ method performs best with $\lambda = 1$. Thus, we fix it and add a new loss term summing over the lifting loss of the third, fourth and fifth layers, which is balanced by a hyperparameter $\mu$ (Note that we divided the Resnet50 into five blocks, meaning that it has five layers in total in our settings. Please refer to Appendix F.1 for more details about the model architecture). Finally the training loss of the experiments testing high-level layers can be formulated as follows:

$$\min_{\delta} L(f_{\theta}, x, \delta) = -\mathbb{E}_{x \sim D_p} \left[ \sum_{i=1}^{2} ||f_{\theta}^i(x + \delta)||_F^2 + \mu \sum_{j=3}^{5} ||f_{\theta}^j(x + \delta)||_F^2 \right], \quad (4)$$

We report the attack success rate (%) of using different $\mu$ against Resnet50 pre-trained by SimCLRv2. Note that C10 stands for CIFAR10, and C100 stands for CIFAR100. Results are shown in the Table 10.

Table 10: The attack success rate(%) of different $\mu$ against **Resnet50** pretrained by **SimCLRv2**. Note that C10 stands for CIFAR10 and C100 stands for CIFAR100.

| $\mu$ | Cars | Pets | Food | DTD | FGVC | CUB | SVHN | C10 | C100 | STL10 | AVG |
|---|---|---|---|---|---|---|---|---|---|---|---|
| 0 | 96.00 | **59.80** | **65.00** | **77.93** | 95.02 | **85.05** | 69.39 | 64.41 | 76.29 | 37.54 | **72.64** |
| 0.01 | 95.32 | 56.99 | 63.60 | 76.06 | 94.93 | 83.31 | **69.65** | 67.30 | 77.55 | **37.76** | 72.25 |
| 0.1 | 95.00 | 56.06 | 62.02 | 75.00 | **95.32** | 82.21 | 66.76 | 70.36 | **77.86** | 35.28 | 71.59 |
| 1 | 95.16 | 56.55 | 62.11 | 74.95 | 95.20 | 82.27 | 67.15 | 69.20 | 78.47 | 35.75 | 71.68 |
| 10 | **96.17** | 55.16 | 62.08 | 77.82 | 95.26 | 83.52 | 66.48 | 67.37 | 77.56 | 33.39 | 71.48 |
| 100 | 61.78 | 42.71 | 59.86 | 70.05 | 86.68 | 66.26 | 63.43 | **86.99** | 64.92 | 21.78 | 62.44 |

As seen from the last column, the larger the weight of the loss of the high-level layers, the worse it performs. Moreover, when the loss of high-level layers overwhelms the low-level ones, the method suffers a significant performance drop (over 10%) in attack success rates. These results show that adding the high-level loss to the training loss bears negative effects.

### C.3  Hyperparameters in the Uniform Gaussian Sampling.

We chose these hyperparameters as $\mu_l = 0.4$, $\mu_h = 0.6$, $\sigma_l = 0.05$, $\sigma_h = 0.1$ in the experiments. The reasons are as follows. For $\mu_l$ and $\mu_h$, we aim to make the mean $\mu$ drawn from $U(\mu_l, \mu_h)$ distributed around 0.5, since the input images are normalized to $[0, 1]$. Thus we tried several configurations of $(\mu_l, \mu_h)$, such as (0.4, 0.6) and (0.45, 0.55), and found that (0.4, 0.6) performs best. For $\sigma_l$ and $\sigma_h$, we hope that most of the samples $n_0 \sim N(\mu, \sigma)$ lie in $[0, 1]$. Thus $\sigma \sim U(\sigma_l, \sigma_h)$ cannot be too large. We also tried some configurations of $(\sigma_l, \sigma_h)$ and found that (0.05, 0.1) performs best.

Interestingly, the set (0.4, 0.6, 0.05, 0.10) generalizes well across the three models. Thus we did not tune these hyperparameters for each model but adopted a single configuration in Table 1, Table 2, and Table 3.

### C.4  Pixel-level perturbations

To test the pixel-level perturbations, we add $\epsilon = 0.05$ to the input images and then evaluate the performance on the three pre-trained models studied in the paper. Then we report the average attack success rate(%) on the ten datasets in Table 11, and detailed performance in Table 12. Note that SimR101, SimR101 and MAEViT stand for Resnet101 pretrained by SimCLRv2, Resnet50 pretrained by SimCLRv2 and ViT-base-16 pretrained by MAE, respectively.

Table 11: The average attack success of different methods against the three models. Note that C10 stands for CIFAR10 and C100 stands for CIFAR100.

| methods | FFF$_{no}$ | FFF$_{mean}$ | FFF$_{one}$ | STD | SSP | ASV | UAP | EPGD | L4A$_{base}$ | L4A$_{fuse}$ | L4A$_{ugs}$ | Pixel |
|---|---|---|---|---|---|---|---|---|---|---|---|---|
| SimR101 | 48.55 | 44.22 | 40.26 | 42.63 | 40.75 | 46.65 | 43.86 | 59.34 | 66.89 | 71.90 | 72.20 | 12.97 |
| SimR50 | 43.86 | 52.10 | 52.98 | 48.26 | 53.64 | 58.19 | 55.16 | 69.28 | 71.16 | 72.64 | 77.80 | 13.76 |
| MAEViT | 77.69 | 71.06 | 74.35 | 27.02 | 53.40 | 22.64 | 55.16 | 66.95 | 94.00 | 94.42 | 95.30 | 12.52 |

Table 12: The attack success rate(%) of the pixel-level attack on ten datasets. Note that C10 stands for CIFAR10 and C100 stands for CIFAR100.

| ASR | Cars | Pets | Food | DTD | FGVC | CUB | SVHN | C10 | C100 | STL10 | AVG |
|---|---|---|---|---|---|---|---|---|---|---|---|
| SimR101 | 10.42 | 9.70 | 12.15 | 29.36 | 23.79 | 21.25 | 2.61 | 2.21 | 15.46 | 2.75 | 12.97 |
| SimR50 | 10.73 | 11.61 | 12.15 | 29.57 | 26.67 | 22.47 | 2.68 | 2.53 | 16.14 | 3.09 | 13.76 |
| MAEViT | 9.92 | 6.90 | 10.53 | 26.28 | 33.75 | 17.96 | 2.64 | 2.50 | 11.99 | 2.76 | 12.52 |

As we can see from the tables, the pixel-level perturbations have little effect on the predictions.

## D  Datasets

We evaluate the performance of pre-trained adversarial perturbations on the CIFAR100 and CIFAR10 [8], STL10 [4], Cars [7], Pets [12], Food [1], DTD [3], FGVC [10], CUB [13], SVHN [11].

We report the calibration (fine-grained or coarse-grained) and the accuracy on clean samples in Table D.

Table 13: Calibration and ACC (%)

| Dataset | Cars | Pets | Food | DTD | FGVC | CUB | SVHN | CIFAR10 | CIFAR100 | STL10 |
|---|---|---|---|---|---|---|---|---|---|---|
| Calibration | fine | fine | fine | fine | fine | fine | fine | coarse | coarse | coarse |
| Resnet101 ACC | 89.80 | 90.60 | 87.90 | 71.01 | 77.04 | 78.78 | 97.40 | 97.85 | 84.81 | 97.33 |
| Resnet50 ACC | 89.35 | 88.20 | 87.84 | 70.60 | 74.01 | 78.13 | 97.40 | 97.51 | 84.03 | 97.00 |
| ViT ACC | 90.03 | 93.48 | 89.60 | 73.60 | 67.16 | 82.32 | 97.38 | 98.10 | 88.03 | 97.20 |

Our datasets do not involve these issues.

# E    Visualisation of Perturbations

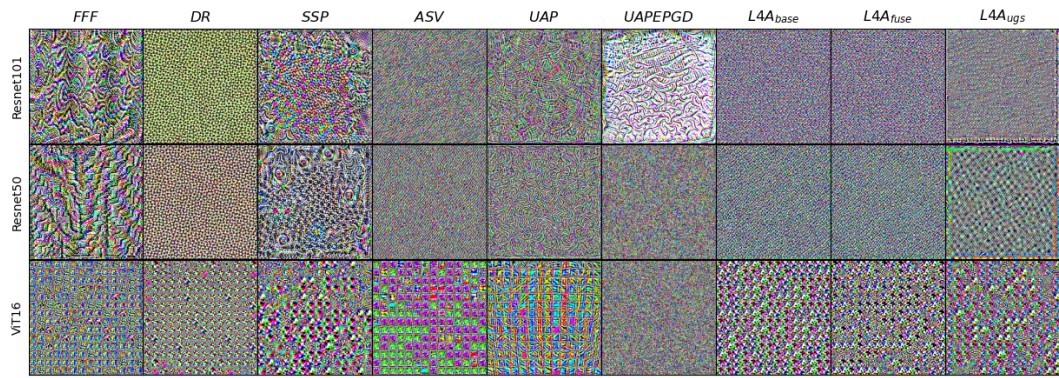

Figure 2: Visualization of pre-trained adversarial perturbations

# F    Implementations

## F.1    Model architecture

To evaluate the effect of attacking different layers, we divide Resnet50, Resnet101 and ViT16 into 5 parts. Here we provide the mapping relationship from the original name to the five layers, respectively.

Table 14: Model architecture

| Layers | layer1 | layer2 | layer3 | layer4 | layer5 |
|---|---|---|---|---|---|
| Resnet50 | net[0] | net[1] | net[2] | net[3] | net[4] |
| Resnet101 | net[0] | net[1] | net[2] | net[3] | net[4] |
| ViT16 | blocks[0] | blocks[1,2,3] | blocks[4,5,6] | blocks[7,8,9] | blocks[10,11] |

## F.2    Resources

We use one Nvidia GeForce RTX 2080 Ti for generating and evaluating PAPs.