# OpenReview forum: "Pre-trained Adversarial Perturbations"
_NeurIPS.cc/2022/Conference — NeurIPS 2022 Accept_

### Official Review · Reviewer_ZvUM · 2022-07-05

**Rating:** 5
**Confidence:** 4
**Soundness:** 2 fair
**Presentation:** 2 fair
**Contribution:** 3 good

**Summary:**

This paper deals with robustness of pretrained models (i.e. self-supervised feature learners which can be used for multiple downstream tasks).  The goal of this paper is to develop "pretrained adversarial perturbations" (short: PAP) -- perturbations that are crafted to attack pretrained models AND their downstream tasks, without having any knowledge about the downstream tasks.  To generate PAPs, a method is developed -- Low-Level Layer Lifting Attack (L4A), which "lifts" the neuron activations of lower-level layers of the neural network.

Experiments are performed on the downstream task of image classification.  Two pretraining methods are used:  SimCLR (with convolutional networks) and MAE (with transformer networks).  ImageNet is used as the pretraining dataset.

**Questions:**

## Questions:
1. Line 125:  "In our experiments, we find the lower the layer is, the better it performs, so we choose the first layer as default, such that k = 1.".  Did you compare with pixel-level perturbations?  i.e. instead of perturbing the feature space, perturb the pixels (this would be equivalent to prior work ike PGD, FGSM etc.). How does PAPs method compare against these?
2. In Table 1, 2 -- UAPEPGD seems to perform significantly better than PAPs for certain datasets (eg. CIFAR10), but on an average PAPs is better -- what might be the reason for this?  Please provide some intuition for this.
3. It is unclear why Sec 4.5 (tradeoff between robustness and clean accuracy) is needed.  This is a well-known result as the paper itself cites Zheng et al for example.  Since the setting of this paper is different, as mentioned by line 250-252, what is the new insight that is being conveyed through Sec 4.5 / Fig 7?

## Suggestions about Formatting, Typos, Structure etc. (Note: these points do not affect my rating, but will be better to address)
1. Please increase the font size for tables 1, 2, 3 and figure 7.
2. Please also include a section on adversarial defenses, and (if there are) techniques developed for defending against trasferable / universal perturbations.
3. Line 114: " To alleviate the lousy effect of the above negative factors" -- the term "lousy" seems out of place here.  I would probably just rephrase it as "To alleviate the negative effect of the above factors".
4. In Table 1, 2, 3, please mention that the % numbers are error rates -- this isn't mentioned anywhere in the paper.


**Limitations:**

Limitations and social impact are mentioned in Sec 6.
This section should also mention that finding from previous sections -- that UAP is a better attacker than PAPs on the pretraining domain -- as a limitation.

**Strengths And Weaknesses:**

## Strengths:
1. Recently, pretrained models have been shown to be useful for many downstream tasks, and in many domains (such as vision, NLP, robotics, audio, ..).  Additionally, "prompt" learning has made pretrained models easy to apply in a zero-shot setting.  Due to these reasons, the idea of transferable adversarial perturbations (i.e. learned on the pretrained model, but also affecting unseen downstream tasks) is interesting and useful for the community.
2. Sec 5 discussion on why UAP is better than PAP on the pretraning domain, is a good contribution.
3. The finding from Fig 2 is interesting -- it shows that lower-level features of the classifier do not deviate much after finetuning on downstream tasks.

## Weaknesses:
1. The proposed method is tested only on image classification tasks.  Often pretrained models are used for many different tasks (for eg. in NLP, pretrained language models can be used for text classification, text generation, machine translation, question answering, etc.  Recently, there has also been an effort to use pretrained vision-language models for many different downstream tasks.  This paper falls short of evaluating the adversarial perturbations on *different* tasks (not just different datasets for the same task).  See:
    i. https://arxiv.org/abs/2204.14198
    ii. https://arxiv.org/abs/2104.00743
2. Eq 6 ($PAP_{UGS}$) method, uses information from the finetuning datasets -- this should be clearly mentioned in the paper.  It only becomes apparent after reading Eq 6.

Overall -- the paper does a good job at defining a new problem statement, providing a method, and showing empirical results on one task (image classification).  However, I find it lacking in terms of transferability to other tasks (see weakness 1).

---

> ### Author Response · Authors · 2022-08-02
> **Thank you for the valuable review (Part 2/2)**
>
> ***Question 7: Include a section on adversarial defenses (if any).***
>
> Thanks for the suggestion. We further consider adversarial fine-tuning by using the method from [1], which adopts a distillation term to preserve high-quality features of the pre-trained model to boost the performance from a view of information theory. We conduct additional experiments on **Resnet50** pretrained by **SimCLRv2** and report the average attack success rate (\%) of different methods against adversarial fine-tuned models. Please refer to Appendix A.2 for detailed performance on the ten datasets. The results are as follows:
>
> | ASR  | FFF$_{\text{no}}$    | FFF$_{\text{mean}}$  | FFF$_{\text{one}}$    | DR       | SSP      | ASV       | UAP       | UAPEPGD  | **L4A$_{\text{base}}$**  | **L4A$_{\text{fuse}}$** | **L4A$_{\text{ugs}}$**   |
> |:---:|:------:|:-------:|:------:|:------:|:------:|:------:|:------:|:-------:|:-------:|:-------:|:------:|
> | **AVG** | 29.74  |  29.64  | 30.04  | 29.74  | 29.59  | 21.95  | 29.38  |  28.77  |  30.46  |  30.80  | 30.04  |
>
> It can be seen from the table that all methods suffer from the degenerated performance against adversarial fine-tuning. However, L4A still performs best among these competitors.
>
> Due to time and space limits, we haven't designed any techniques for defending against transferable / universal perturbations. We leave it for feature work.
>
>
> ***Question 8: Other minor issues.***
>
> Thanks for pointing them out. We have fixed these problems in the revision. We will further improve the paper in the final.
>
> **Reference:**
>
> [1] Dong et al. How Should Pre-Trained Language Models Be Fine-Tuned Towards Adversarial Robustness? NeurIPS 2021.

---

> > ### Comment · Reviewer_ZvUM · 2022-08-07
> > **Response to Rebuttal**
> >
> > Thanks for the detailed rebuttal.  Here are my thoughts:
> >
> > **Question 1: The proposed method is tested only on image classification tasks.**
> > I did not mean that this paper should test on NLP / V&L tasks.  What I meant was that pretrained adversarial perturbations should be investigated not just on image classification tasks, but also on other "vision" tasks that have adversarial attack/defense benchmarks.
> >
> > **Question 2:  method uses information from the finetuning datasets.**
> > Thanks for the clarification.  How were the hyperparameters $\mu_l, \mu_h, \sigma_l, \sigma_h$ chosen? (line 143-144 in the updated pdf)
> >
> > **Question 3: Did you compare with pixel-level perturbations?**
> > I'm not convinced by this response.  I'm looking for an answer to the following question:
> > - in the paper, you have stated that _"the lower the layer is, the better it performs"_ and you chose k=1.
> > - what if you go even lower (i.e. at the pixel level) and apply your same method at pixel level?
> >
> > **Q4** -- thanks for the insight w.r.t. size of images per class
> >
> > **Q5 Sec 4.5** -- my point is that this tradeoff is already a well known result.  I appreciate that you have shown that the trade-off is also observed in your pretraining/finetuning setting.  This is valuable, but not surprising.  Is the same tradeoff also observed for PDG/previous forms of adversarial perturbations?  If yes, then you should also add that in Fig 7.

---

> > > ### Author Response · Authors · 2022-08-09
> > > **Thank you for the further feedback (Part 2/2)**
> > >
> > > ***Question 3: Did you compare with pixel-level perturbations?***
> > >
> > > Thanks for the suggestion. As shown in Eq. (4), if we apply our method at the pixel level, we actually minimize $-\mathbb{E}_{x\sim D_p}[||x+\delta||_F^2]$. Since the input images $x$ are constrained in [0,1] as in Eq. (1), this problem has a closed-form solution as $\delta=\epsilon$ under the L_inf norm bound. Based on this, we compare with a baseline that adds $\epsilon=0.05$ to the input images and test the performance of three pre-trained models studied in the paper -- Resnet101 pre-trained by SimCLRv2, Resnet50 pre-trained by SimCLRv2, and ViT16 pre-trained by MAE.  We report the average attack success rate (\%) on the ten datasets in the following table. Please refer to Appendix C.4 for detailed performance on the ten datasets.
> > >
> > > |   ASR   |  FFF$_{\text{no}}$ | FFF$_{\text{mean}}$| FFF$_{\text{one}}$ |   DR   |   SSP  |   ASV  |   UAP  | UAPEPGD | **L4A$_{\text{base}}$** |  **L4A$_{\text{fuse}}$** |  **L4A$_{\text{ugs}}$** | *Pixel-level* |
> > > |:--------:|:------:|:-------:|:------:|:------:|:------:|:------:|:------:|:-------:|:-------:|:-------:|:------:|:--------:|
> > > | Resnet101 (SimCLRv2) | 48.55  |  44.22  | 40.26  | 42.63  | 40.75  | 46.65  | 43.86  |  59.34  |  66.89  |  71.90  | 72.20  |  *12.97*   |
> > > |  Resnet50 (SimCLRv2) | 43.86  |  52.10  | 52.98  | 48.26  | 53.64  | 58.19  | 55.16  |  69.28  |  71.16  |  72.64  | 77.80  |  *13.76*   |
> > > |  ViT16 (MAE) | 77.69  |  71.06  | 74.35  | 27.02  | 53.40  | 22.64  | 55.16  |  66.95  |  94.00  |  94.42  | 95.30  |  *12.52*   |
> > >
> > > As can be seen from the table, the pixel-level perturbations have little effect on the predictions, leading to much lower attack success rates.
> > >
> > > ***Question 5: Is the same tradeoff also observed for PDG/previous forms of adversarial perturbations?***
> > >
> > > Thanks for pointing it out. We agree that the trade-off between accuracy and robustness is a well-known result. We also think that it is valuable to study this trade-off in the pre-training/fine-tuning setting. The major difference between the findings of previous works and ours is that they mainly consider model robustness under white-box attacks, while we consider model robustness under transfer-based PAPs. Since the fine-tuned models are not adversarially trained, they can hardly preserve adversarial robustness under stronger white-box attacks (such as PGD). We found that their white-box robustness under PGD is 0\% along the whole process of fine-tuning. Therefore, to show the robustness curve as in Fig. 7, we adopt the weaker FGSM attack. We found that the model robustness under the white-box FGSM attack tends to plateau after a certain epoch and does not show the trade-off phenomenon. We think that along fine-tuning, model robustness under white-box attacks tends to maintain, but the robustness under PAPs increases since model parameters differs more from the pre-trained models.
> > > In the new revision, we add this result in Fig. 7 as suggested.
> > >
> > > We are sorry to provide the response a little bit late due to the time of running additional experiments, especially the experiments on object detection and semantic segmentation. We hope that our new results can address your concerns and you can find our response satisfactory. We will further improve the paper in the final version.

---

> > > ### Author Response · Authors · 2022-08-09
> > > **Thank you for the further feedback (Part 1/2)**
> > >
> > > Thank you for the follow-up comments. We have conducted more experiments to address your concerns and also uploaded a new revision of our paper. Below we provide the detailed responses.
> > >
> > > ***Question 1: The proposed method is tested only on image classification tasks.***
> > >
> > > Thanks for the suggestion. We conduct additional experiments on the object detection and semantic segmentation tasks to evaluate the performance of our method.
> > >
> > > For object detection, we adopt the off-the-shelf Resnet50 model that is pre-trained by MOCOv2 on the ImageNet dataset and fine-tuned on the COCO dataset (the model is available at MMDetection). We generate the pre-trained adversarial perturbations (PAPs) against the pre-trained model and then evaluate the performance against the fine-tuned model on the COCO validation set. The evaluation metrics include mAP (\%), mAP$_\text{50}$ (\%), and mAP$_\text{75}$ (\%).
> > > The results are as follows:
> > >
> > > | method | FFF$_{\text{no}}$ | FFF$_{\text{mean}}$| FFF$_{\text{one}}$ |   DR  |  SSP  |  ASV  |  UAP  |  UAPEPGD |  **L4A$_{\text{base}}$** |  **L4A$_{\text{fuse}}$** |  **L4A$_{\text{ugs}}$**  |
> > > |:-----------:|:-------:|:--------:|:----------:|:------:|:------:|:------:|:------:|:-------:|:------:|:------:|:-------:|
> > > |   mAP  |  30.7  |  30.0   |   30.8    | 31.6  | 31.0  | 29.8  | 30.2  |  34.2  | 29.8  | 29.3  |  **26.5**  |
> > > | mAP$_\text{50}$  |  48.5  |  47.6   |   48.6    | 49.6  | 48.6  | 46.9  | 47.8  |  53.1  | 46.9  | 46.2  |  **42.5**  |
> > > | mAP$_\text{75}$  |  32.9  |  32.1   |   33.0    | 34.2  | 33.4  | 31.9  | 32.5  |  37.4  | 32.0  | 31.6  |  **28.2**  |
> > >
> > > It shows that our proposed methods outperform all the baselines by a large margin. For example, the best competitor ASV achieves mAP$_\text{50}$ of **46.9\%**, while our method L4A$_\text{ugs}$ degrades mAP$_\text{50}$ to **42.5\%**, showing the effectiveness.
> > >
> > > For semantic segmentation, we adopt a ViT-based model that is pre-trained by MAE on the ImageNet dataset and fine-tuned on the ADE20k dataset (the model is available at MMSegmentation).
> > > We evaluate the performance of different methods under the metric of mIoU (\%). The results are as follows:
> > >
> > > | method | FFF$_{\text{no}}$ | FFF$_{\text{mean}}$| FFF$_{\text{one}}$ |   DR  |  SSP  |  ASV  |  UAP  |  UAPEPGD |  **L4A$_{\text{base}}$** |  **L4A$_{\text{fuse}}$** |  **L4A$_{\text{ugs}}$**  |
> > > |:------:|:-------:|:--------:|:----------:|:-----:|:-----:|:-----:|:-----:|:-----:|:-----:|:-----:|:-----:|
> > > |  mIoU  |  40.63  |   40.79  |    40.86   | 42.67 | 41.99 | 41.84 | 41.89 | 41.07 | **39.38** | 39.59 | 39.44 |
> > >
> > > From the table, the best competitor $FFF_{\text{no}}$ achieves mIoU of **40.63\%**, while $L4A_{\text{base}}$ degrades mIoU to **39.38\%**, which is also more effective.
> > > The results on object detection and semantic segmentation show that PAPs can generalize well to other vision tasks, demonstrating the good cross-task transferability of the generated perturbations. In the new revision, we add the results in Appendix A.4.
> > >
> > > ***Question 2: How were the hyperparameters $\mu_{l}$, $\mu_{h}$, $\sigma_{l}$, $\sigma_{h}$ chosen?***
> > >
> > > We chose these hyperparameters as $\mu_l=0.4$, $\mu_h=0.6$, $\sigma_l=0.05$, $\sigma_h=0.1$ in the experiments. The reasons are as follows.
> > > For $\mu_l$ and $\mu_h$, we aim to make the mean $\mu$ drawn from $U(\mu_l,\mu_h)$ distributed around 0.5, since the input images are normalized to $[0,1]$. Thus we tried several configurations of $(\mu_l,\mu_h)$, such as (0.4, 0.6) and (0.45, 0.55), and found that (0.4, 0.6) performs best.
> > > For $\sigma_{l}$ and $\sigma_{h}$, we hope that most of the samples $n_0\sim N(\mu, \sigma)$ lie in $[0,1]$. Thus $\sigma\sim U(\sigma_l,\sigma_h)$ cannot be too large. We also tried some configurations of ($\sigma_{l}$, $\sigma_{h}$) and found that (0.05, 0.1) performs best.
> > >
> > > Moreover, the setting of these hyperparameters as (0.4, 0.6, 0.05, 0.1) generalizes well across all three models in the experiments. Thus we did not tune these hyperparameters for each model but adopted a single configuration in Table 1, Table 2, and Table 3. We clarify this in Appendix C.3 in the revision.

---

> ### Author Response · Authors · 2022-08-02
> **Thank you for the valuable review (Part 1/2)**
>
> Thank you for the valuable review. We have uploaded a revision of our paper. Below we address the detailed comments.
>
> ***Question 1: The proposed method is tested only on image classification tasks.***
>
> We agree that there are many tasks related to pre-trained models in the fields of computer vision, natural language processing, etc. Since vision and language are extremely different domains that vision tasks process continuous images while natural language processing tasks deal with discrete words, the attack methods are also different in these two domains. Thus our method cannot be simply extended to pre-trained language models. However, its basic idea may be general among them. For vision-and-language models, we further conduct experiments on the pre-trained CLIP model to show the effectiveness of our method (see Appendix A.1 in the revision). We will further consider more vision-and-language pre-trained models in the final.
>
> ***Question 2: $L4A_{\text{ugs}}$ method uses information from the finetuning datasets.***
>
> We want to clarify that we do not have access to the fine-tuning datasets and the fine-tuned models. Thus, we haven't used any information from the fine-tuning datasets in our methods including $L4A_{\text{ugs}}$.
> The mean and variance used to generate the Gaussian noise in $L4A_{\text{ugs}}$ are sampled from uniform distributions. The motivation is that the uniformly sampled mean and variance would cover those of the downstream tasks.
>
> ***Question 3: Did you compare with pixel-level perturbations?***
>
> Our PAPs are pixel-level perturbations -- we add the PAPs to the input images to fool the model. Our method also adopts the projected gradient descent method to generate PAPs based on the loss defined on low-level layers. In fact, we have compared our method with an enhanced PGD method--UAPEPGD, which adds a momentum term to boost the black-box transferability. The main difference between our method and conventional PGD-based methods is that we design loss functions on low-level layers, while they adopt loss based on high-level layers.
>
>
> ***Question 4: UAPEPGD seems to perform significantly better than PAPs for certain datasets. (eg. CIFAR10)***
>
> There are mainly two reasons.
> Firstly, as CIFAR10 is a coarse-grained dataset, the model tends to use various high-level features to complete the task. Thus directly optimizing the high-level classification scores in UAPEPGD may help. Meanwhile, in a coarse-grained classification task, more features are used. So, there are more adversarial directions for an input. From this perspective, the low-gradient-alignment of UAP and UAPEPGD (Please refer to Section 5 for details about gradient alignment) allows them to choose the adversarial directions more flexibly and update in the quickest one.
> Secondly, the CIFAR10 dataset bears a large capacity. The number of images per class in CIFAR10 is nearly ten times that of CIFAR100 and STL10. When fine-tuning the model, more features can be learned. Then it is the same as the former explanation -- the more features, the better UAPEPGD performs. Another piece of evidence is that UAPEPGD performs well on Food: The number of images per class in Food is the largest in the seven fine-grained datasets.
>
> ***Question 5: It is unclear why Sec 4.5 (tradeoff between robustness and clean accuracy) is needed.***
>
> Previous works mainly focus on the effect of training settings (such as the attack method in adversarial training and the hyperparameters) on the tradeoff. However, our work emphasizes the dynamic behaviour of the fine-tuning procedure. Instead of training models in different settings, we just fine-tune one model and monitor the curves of clean accuracy and robustness during the fine-tuning procedure. In Figure 5, as the training epoch increases, the clean accuracy is monotonically rising, while the robustness shows a trend from increase to decrease. This may reveal that the fine-tuned models tend to use spurious features rather than the robust features learned during pre-training. Furthermore, the fine-tuned models can forget useful features even if they have learned them, as the robustness curve tends to decline after achieving a peak. These findings reveal the vulnerability of the standard fine-tuning procedure, and more work should be done to help the model utilize more robust features.
>
> ***Question 6: The figures and tables are two small.***
>
> Thanks for the suggestion. In the revision, we adjust the size of the figures and tables.

---

### Official Review · Reviewer_qc7g · 2022-07-10

**Rating:** 6
**Confidence:** 2
**Soundness:** 3 good
**Presentation:** 2 fair
**Contribution:** 3 good

**Summary:**

This paper aims to propose a new attack method that can specifically improve the attack success rate on big pre-trained models. The newly proposed two techniques, Low-Level Layer Lifting Attack and Uniform Gaussian Sampling, have demonstrated notable performance improvement.

**Questions:**

Please refer to my previous questions on the weakness part.

**Limitations:**

Yes

**Strengths And Weaknesses:**

## Strengths

(1) The proposed methods sound reasonable. Damaging the low-level features of the pre-trained models intuitively should have the ability to attack the downstream tasks.

(2) The experimental results have demonstrated the effectiveness of the proposed methods, outperforming the previous methods by a large margin.

(3) The analysis of the proposed method is quite extensive, almost covering all the aspects I can think of. The trade-off between the clean accuracy and robustness is interesting.

## Weaknesses

(1) All the figures and tables are two small. I need to zoom in by 300% to clearly see the contents.

(2) I am curious about the formulation of Eq4. Why don't we maximize ||f(x + δ) - f(x)||, but to maximize -f(x + δ)?

---

> ### Author Response · Authors · 2022-08-02
> **Thank you for the supportive review**
>
> Thank you for appreciating our new contributions as well as providing the valuable feedback. We have uploaded a revision of our paper. Below we address the detailed comments.
>
> ***Question 1: All the figures and tables are two small.***
>
> Thanks for the suggestion. In the revision, we adjust the size of the figures and tables to make them clearer.
>
>
> ***Question 2:  Why don't we maximize $||f(x + \delta) - f(x)||_2^{2}$, but to maximize $||f(x + \delta)||_2^{2}$ in Eq. (4)?***
>
> In fact, the SSP method tries to maximize $||f(x + \delta) - f(x)||_2^{2}$ and it suffers from a drop in attack success rate compared to maximizing $||f(x + \delta)||_2^{2}$ by our methods.
> Considering the gradient-alignment theory in Section 5., we think that maximizing $||f(x + \delta)||_2^{2}$ tends to bear high gradient-alignment. The intuition is as follows. Maximizing $||f(x + \delta) - f(x)||_2^{2}$ tries to make the representations of the clean and the adversarial inputs as far as possible in the feature space. Their feature representations can vary a lot for different inputs, so the perturbations obtained by computing the distance may be far from each other. However, if we maximize $||f(x + \delta)||_2^{2}$, this is less concerned with the specific $x$. We just try to activate all the neurons in the low-level layers to mask the useful features, regardless of what the useful features (and the inputs) are. For example, the optimal objective of $||f(x + \delta)||_2^{2}$ can be obtained by setting the values of the activation to the maximum for every $x$.

---

### Official Review · Reviewer_2UeR · 2022-07-11

**Rating:** 5
**Confidence:** 4
**Soundness:** 1 poor
**Presentation:** 2 fair
**Contribution:** 1 poor

**Summary:**

This work proposes a new type of adversarial attack that aims at the pre-trained models, which could still fool the pre-trained model after fine-tuning. The proposed method is built upon the observation that the lower layers of the pre-trained model hardly change after fine-tuning. Specifically, they propose to generate a perturbation that can fool the feature of lower layers. Empirical results demonstrate a high attack success rate after fine-tuning.

**Questions:**

My concerns and questions are listed in the weakness part.

**Limitations:**

No social negative impact is found here.

**Strengths And Weaknesses:**

Strengths
1. Figure 2 nicely shows the feature in the lower layers only change a little.
2. The performance on multiple datasets is reported.

Weaknesses
1. For the motivation, I am wondering if this setting is realistic. In practice, it is way harder to get the pre-trained model than the weight of the model. Moreover, knowing the weight of the model can make the adversarial attack much more successful (since we can attack both low and deep layers). In this case, why do we still want to attack the pre-trained model?

2. For evaluation, the authors only attack the standard pre-trained models. I think it is necessary for conducting experiments on the adversarial pre-trained models, which are obviously more robust than the standard pre-trained models.

3. For evaluation, the fine-tuning setting is not described. Which fine-tuning is performed here, adversarial or standard fine-tuning? If it's adversarial fine-tuning, which adversarial fine-tuning method is employed here? If it's standard fine-tuning, what's the attack success rate for adversarial fine-tuned models.

4. There is no detailed description in the caption of Table 1, Table 2, and Table 3. It is hard to learn the meaning of the percentage value in these Tables

---

> ### Author Response · Authors · 2022-08-02
> **Thank you for the valuable review**
>
> Thank you for the valuable review. We have uploaded a revision of our paper. Below we address the detailed comments, and hope that you can find our response satisfactory.
>
> ***Question 1: Why do we attack the pre-trained models?***
>
> In this paper, we consider the scenario that the pre-trained models are publicly available and the attacker aims to generate adversarial perturbations before the model has been fine-tuned on downstream tasks. This setting is practical in the field of pre-training. Specifically, many large-scale pre-trained models are released or at least provide API access. The users can then fine-tune these models to do any downstream tasks. When deploying the fine-tuned models to real applications, these models are usually not released as black-box models to the attacker. Therefore, the attacker aims to pre-generate the adversarial perturbations with the purpose of remaining effective for any (unknown) downstream task. To this end, we propose a novel method to generate pre-trained adversarial perturbations that correspond to this setting.
>
> ***Question 2: Experiments on the adversarial pre-trained models.***
>
> We survey the existing papers on adversarial pre-training in vision tasks and find that all works perform pre-training on small datasets, such as CIFAR or ImageNet-200 (a subset of ImageNet). Most of the fine-tuned models are only evaluated on the same dataset. However, our work mainly focuses on large-scale pre-trained models on ImageNet, which learn many useful features and can be easily fine-tuned to many downstream tasks.
> Besides, there are large-scale adversarial pre-trained vision-and-language models [1]. However, their tasks, such as VQA, are very different from ours and there exist no prior works on universal adversarial perturbations in vision-and-language tasks.
> Therefore, we are unable to conduct experiments on adversarial pre-trained models for now. We will further conduct this experiment by performing adversarial pre-training on ImageNet in the final version.
>
> ***Question 3: What's the attack success rate for adversarial fine-tuned models.***
>
> Thanks for the suggestion.
> In our paper, we only conducted experiments on standard fine-tuning. We further consider adversarial fine-tuning by using the method from [2], which adopts a distillation term to preserve high-quality features of the pre-trained model to boost the performance from a view of information theory. We conduct some additional experiments on **Resnet50** pretrained by **SimCLRv2** and report the average attack success rate (\%) of different methods against adversarial fine-tuned models. Please refer to the Appendix A.2 for detailed performance on the ten datasets. The results are as follows:
>
> | ASR  | FFF$_{\text{no}}$    | FFF$_{\text{mean}}$  | FFF$_{\text{one}}$    | DR       | SSP      | ASV       | UAP       | UAPEPGD  | **L4A$_{\text{base}}$**  | **L4A$_{\text{fuse}}$** | **L4A$_{\text{ugs}}$**   |
> |:---:|:------:|:-------:|:------:|:------:|:------:|:------:|:------:|:-------:|:-------:|:-------:|:------:|
> | **AVG** | 29.74  |  29.64  | 30.04  | 29.74  | 29.59  | 21.95  | 29.38  |  28.77  |  30.46  |  30.80  | 30.04  |
>
> It can be seen from the table that all methods suffer from the degenerated performance against adversarial fine-tuning. However, $L4A$ still performs best among these competitors.
>
> ***Question 4: Lack of details in the captions of the tables.***
>
> Thanks for pointing this out. In the revision, we provide the detailed description of the tables in the caption. We will further improve the writing in the final.
>
> **Reference:**
>
> [1] Radford et al. Learning Transferable Visual Models from Natural Language Supervision. ICML 2021.
>
> [2] Dong et al. How Should Pre-Trained Language Models Be Fine-Tuned Towards Adversarial Robustness? NeurIPS 2021.

---

> > ### Comment · Reviewer_2UeR · 2022-08-05
> > **Response to authors**
> >
> > I am not convinced by the reason why experiments on adversarial pre-training are missing: I think models pre-trained on ImageNet-200 are elegant enough to serve as pre-train.

---

> > > ### Author Response · Authors · 2022-08-08
> > > **Thank you for the further feedback**
> > >
> > > Thank you for the follow-up comment. As suggested, we further consider adversarial pre-training on ImageNet-200. Following Jiang et al. [1], we pre-train a Resnet50 model by adversarial contrastive learning. Then, we generate our pre-trained adversarial perturbations against this model and test the performance on downstream tasks. For fine-tuning, we consider both standard fine-tuning and adversarial fine-tuning [2]. The following table shows the average attack success rate (\%) on the ten datasets against standard fine-tuned models. Please refer to Appendix A.3 for detailed performance on the ten datasets.
> > >
> > >
> > > | ASR  | FFF$_{\text{no}}$    | FFF$_{\text{mean}}$  | FFF$_{\text{one}}$    | DR       | SSP      | ASV       | UAP       | UAPEPGD  | **L4A$_{\text{base}}$**  | **L4A$_{\text{fuse}}$** | **L4A$_{\text{ugs}}$**   |
> > > |:---:|:------:|:-------:|:------:|:------:|:------:|:------:|:------:|:-------:|:-------:|:-------:|:------:|
> > > | **AVG** | 53.61  |  59.45  | 53.33  | 55.69  | 54.21  | 53.09  | 53.84  |  34.33  |  63.47  |  65.85  | **68.91**  |
> > >
> > > The above table shows that standard fine-tuning can hardly preserve the robustness of adversarial pre-trained models, which is also reported in [3,4].
> > > In such a setting, L4A performs best among all competitors: the best baseline $FFF_{\text{mean}}$ achieves an average attack success rate of **59.45\%**, while that of the villain $L4A_{\text{base}}$ is up to **63.47\%**, and the uniform Gaussian sampling further boosts the performance to **68.91\%**.
> > >
> > > Furthermore, we report the average attack success rate (\%) of the ten downstream tasks against adversarial fine-tuned models in the following table. Please refer to Appendix A.3 for detailed performance on the ten datasets.
> > >
> > > | ASR  | FFF$_{\text{no}}$    | FFF$_{\text{mean}}$  | FFF$_{\text{one}}$    | DR       | SSP      | ASV       | UAP       | UAPEPGD  | **L4A$_{\text{base}}$**  | **L4A$_{\text{fuse}}$** | **L4A$_{\text{ugs}}$**   |
> > > |:---:|:------:|:-------:|:------:|:------:|:------:|:------:|:------:|:-------:|:-------:|:-------:|:------:|
> > > | **AVG** | 30.18   |  30.23  | 30.12  | 30.36  | 29.03  |  29.52  |  28.75   |  27.20  |  31.49  | 31.38  | **31.60**|
> > >
> > > We can see that adversarial fine-tuning leads to better robustness than standard fine-tuning, which is consistent with Appendix A.2 and the finding in [3] that adversarial fine-tuning contributes to the final robustness more than adversarial pre-training.
> > > Although all methods achieve relatively low attack success rates, L4A is still the best among them.
> > >
> > > We hope that our new results can address your concerns. We will further improve the paper in the final version.
> > >
> > > Reference:
> > >
> > > [1] Jiang, Ziyu, et al. "Robust pre-training by adversarial contrastive learning." NeurIPS 2020.
> > >
> > > [2] Dong, Xinshuai, et al. "How Should Pre-Trained Language Models Be Fine-Tuned Towards Adversarial Robustness?." NeurIPS 2021.
> > >
> > > [3] Chen, Tianlong, et al. "Adversarial robustness: From self-supervised pre-training to fine-tuning." CVPR 2020.
> > >
> > > [4] Kumar, Ananya, et al. "Fine-tuning can distort pretrained features and underperform out-of-distribution." arXiv preprint arXiv:2202.10054 (2022).

---

> > > > ### Comment · Reviewer_2UeR · 2022-08-08
> > > > **The authors have addressed my concerns**
> > > >
> > > > Thanks for the timely and detailed response from the authors. My concerns have been addressed and I would raise the score.

---

> > > > > ### Author Response · Authors · 2022-08-09
> > > > > **Thanks for the update**
> > > > >
> > > > > Thank you very much for the increase on rating and valuable feedback. We really appreciate that. We'll try our best to further improve this paper in the final version.

---

### Official Review · Reviewer_z1MX · 2022-07-13

**Rating:** 6
**Confidence:** 3
**Ethics Flag:** Yes
**Soundness:** 2 fair
**Presentation:** 3 good
**Contribution:** 3 good

**Summary:**

This paper tries to attack the pre-trained models (when handling the downstream tasks) by introducing a Low-Level Layer Lifting Attack (L4A) method, which mainly perturbation the neurons lying at the low-level layers of the pre-trained models.

**Questions:**

For the `Low-Level Layer Lifting Attack`, the authors chose the first and second layers of the pre-trained to attack since they found the layers laying at a higher level, the more their parameters change during fine-tuning. However, this does not mean that attacks the parameters at the higher level layers are useless. Hence, some additional experiments can be done. For example, attack all the parameters at different layers but with some hyperparameters (i.e., loss weights to balance each term). Also, I think there should be a hyperparameter in `Eq.5` between those two norm.

**Ethics Review Area:**

["Privacy and Security (e.g., consent)"]

**Limitations:**

Yes

**Strengths And Weaknesses:**

Strengths:

The method looks simple and effective. It greatly improved the attack success rate compared to the SOTA.


Weaknesses:

1. Figure 1 gives little information about this method, and I think more detailed information should be given in the caption.
2. The author just uses three pre-trained models, two of which are similar (ResNet50 and ResNet101); how about the performance on CLIP or the other pre-trained models?

---

> ### Author Response · Authors · 2022-08-02
> **Thank you for the valuable review**
>
> Thank you for the valuable review. We have uploaded a revision of our paper. Below we address the detailed comments.
>
> ***Question 1: Figure 1 gives little information about this method, and more detailed information should be given in the caption.***
>
> Thanks for the suggestion. In the revision, we provide a more detailed description of Figure 1 in the caption.
>
> ***Question 2: How about the performance on CLIP or the other pre-trained models.***
>
> Thanks for the suggestion. In the revision, we conduct additional experiments on CLIP and MOCO. We adopt the pre-trained weights with the Resnet50 backbone.
> Here, we report the average attack success rate (\%) on the ten datasets against MOCO and CLIP below. Please check Appendix A.1 for the detailed performance on the ten datasets.
> | ASR  | FFF$_{\text{no}}$    | FFF$_{\text{mean}}$  | FFF$_{\text{one}}$    | DR       | SSP      | ASV       | UAP       | UAPEPGD  | **L4A$_{\text{base}}$**  | **L4A$_{\text{fuse}}$** | **L4A$_{\text{ugs}}$**   |
> |:------:|:----------:|:----------:|:-----------:|:----------:|:----------:|:-----------:|:-----------:|:----------:|:----------:|:---------:|:----------:|
> | **CLIP** | 89.06 | 89.03   | 86.98 | 77.86 | 86.36  | 88.78  | 81.04 | 73.59 | 81.00 | 91.41 | **93.61** |
> | **MOCO** | 40.30  | 45.99 | 38.59   | 44.92 | 48.42 | 43.83 | 55.34   | 41.89 | 53.94  | 54.72 | **59.74** |
>
> These results show that our proposed methods are also effective on other pre-trained models. For MOCO, the best baseline UAP achieves an average attack success rate of 55.34%, while our method $L4A_{\text{ugs}}$ improves the performance to **59.74%**. For CLIP, the average attack success rate of the best baseline ASV is 88.78%, while $L4A_{\text{ugs}}$ achieves **93.61%** attack success rate on average. The results demonstrate the effectiveness of our method.
>
> ***Question 3: Missing experiments on fusing high-level layers.***
>
> We are sorry that we missed a hyperparameter $\lambda$ in Eq. (5) and Eq. (6) between the two norms, and we have corrected them in the revision. In fact, we conducted an ablation study to test different $\lambda$ in Appendix C.1.
> To study the effect of utilizing the high-level features, we choose Resnet50 pre-trained by SimCLRv2 as the target. In previous experiments, we found that the $L4A_{\text{fuse}}$ method performs best with $\lambda=1$. Thus, we fix it and add a new loss term summing over the lifting loss of the third, fourth and fifth layers, which is balanced by a hyperparameter **$\mu$** (Note that we divided the Resnet50 into five blocks, meaning that it has five layers in total in our settings. Please refer to Appendix F.1 for more details about the model).
>
> We report the average attack success rate (\%) of using different **$\mu$** against **Resnet50** pre-trained by **SimCLRv2** below. Please refer to Appendix C.2 for detailed results on the ten datasets.
>
> | $\mu$ |    0   |  0.01  |   0.1  |    1   |   10   |   100  |
> |:-------:|:------:|:------:|:------:|:------:|:------:|:------:|
> |   AVG   | 72.64  | 72.25  | 71.59  | 71.68  | 71.48  | 62.44  |
>
> As seen in the table, the larger the weight of the loss of the high-level layers, the worse it performs. Moreover, when the loss of high-level layers overwhelms the low-level ones, the method suffers from a significant performance drop (over 10%) in attack success rate. These results show that the high-level layers  bear negative effects when added to the training loss.

---

> ### Comment · Reviewer_z1MX · 2022-08-09
> **Update**
>
> All of my concerns are addressed, and I will raise my score from 5 to 6.

---

> > ### Author Response · Authors · 2022-08-09
> > **Thanks for the update**
> >
> > We really appreciate your increasing rating and valuable comments. We'll spare no effort to improve this paper in the final version. Thank you again.

---

### Review · Ethics_Reviewer_PhYp · 2022-08-12

**Recommendation:**

Could use clarity on what ethical concern the reviewer is flagging, and authors should provide the required social/ethical review statement and questionnaire responses.

**Ethics Review:**

The paper was flagged by a reviewer for "Privacy and Security (e.g., consent)" concerns, but upon reading the manuscript it is unclear to me what ethical concerns exist. The paper does not appear to be focusing on human faces or other issues that might raise issues of privacy or consent. (I am willing to be corrected if mistaken).

That said, the paper lacks any social/ethical statement, nor does it have the submission questionnaire attached. So it is hard to assess if/how the authors have considered any relevant issues.

---

### Author Response · Authors · 2022-08-05
**Look forward to further feedback**

Dear reviewers,

We thank you again for the valuable comments. We are looking forward to hearing from you about any further feedback.

If you find the response satisfactory, we hope you might view this as a sufficient reason to further raise your score.

If you still have questions about our paper, we are willing to discuss them with you and improve our paper.

Best, Authors

---

### Meta-Review · Area_Chair_1w9e · 2022-08-29

**Recommendation:** Accept
**Confidence:** Certain

**Metareview:**

This paper develops an attack method for the pre-trained models so the attack can remain effective even for downstream tasks. The authors introduced a Low-Level Layer Lifting Attack (L4A) method, which mainly perturbation the neurons lying at the low-level layers of the pre-trained models. Their method looks simple and effective, and multiple datasets and settings are reported to justify its effectiveness. During rebuttal, the authors also reported additional results on adversarial pre-training, which would be valuable to add to the final paper.

On the negative side, the proposed method is tested only on the image classification task.  Besides, the authors should have provided the required social/ethical review statement and questionnaire responses - please make sure to add.

Overall, this paper passes the bar given the generally positive sentiment among reviewers.


**Award:**

No

---

### Decision · Program_Chairs · 2022-09-14

Accept